# The long non-coding RNA *Cerox1* is a post transcriptional regulator of mitochondrial complex I catalytic activity

**Tamara M Sirey[1,2]\*, Kenny Roberts[2], Wilfried Haerty[2†], Oscar Bedoya-Reina[1,2‡], Sebastian Rogatti-Granados[1,2], Jennifer Y Tan[2§], Nick Li[2], Lisa C Heather[3], Roderick N Carter[4], Sarah Cooper[5#], Andrew J Finch[1], Jimi Wills[1], Nicholas M Morton[4], Ana Claudia Marques[2§], Chris P Ponting[1,2]\***

[1]MRC Human Genetics Unit, Institute of Genetics and Molecular Medicine, University of Edinburgh, Western General Hospital, Edinburgh, United Kingdom; [2]MRC Functional Genomics Unit, University of Oxford, Oxford, United Kingdom; [3]Department of Physiology, Anatomy and Genetics, University of Oxford, Oxford, United Kingdom; [4]University/British Heart Foundation Centre for Cardiovascular Science, Queen's Medical Research Institute, University of Edinburgh, Edinburgh, United Kingdom; [5]Department of Biochemistry, University of Oxford, Oxford, United Kingdom

**\*For correspondence:**
tamara.sirey@igmm.ed.ac.uk (TMS);
Chris.Ponting@igmm.ed.ac.uk (CPP)

**Present address:** [†]Earlham Institute, Norwich, United Kingdom; [‡]Department of Microbiology, Tumor and Cell Biology, Karolinska Institutet, Solna, Sweden; [§]Department of Computational Biology, University of Lausanne, Lausanne, Switzerland; [#]Wellcome Sanger Institute, Cambridge, United Kingdom

**Abstract** To generate energy efficiently, the cell is uniquely challenged to co-ordinate the abundance of electron transport chain protein subunits expressed from both nuclear and mitochondrial genomes. How an effective stoichiometry of this many constituent subunits is co-ordinated post-transcriptionally remains poorly understood. Here we show that *Cerox1*, an unusually abundant cytoplasmic long noncoding RNA (lncRNA), modulates the levels of mitochondrial complex I subunit transcripts in a manner that requires binding to microRNA-488-3p. Increased abundance of *Cerox1* cooperatively elevates complex I subunit protein abundance and enzymatic activity, decreases reactive oxygen species production, and protects against the complex I inhibitor rotenone. *Cerox1* function is conserved across placental mammals: human and mouse orthologues effectively modulate complex I enzymatic activity in mouse and human cells, respectively. *Cerox1* is the first lncRNA demonstrated, to our knowledge, to regulate mitochondrial oxidative phosphorylation and, with miR-488-3p, represent novel targets for the modulation of complex I activity.
DOI: https://doi.org/10.7554/eLife.45051.001

## Introduction

In eukaryotes, coupling of the mitochondrial electron transport chain to oxidative phosphorylation (OXPHOS) generates the majority of ATP that fulfils cellular energy requirements. The first enzyme of the electron transport chain, NADH:ubiquinone oxidoreductase (complex I), catalyses the transfer of electrons from NADH to coenzyme Q10, pumps protons across the inner mitochondrial membrane and produces reactive oxygen species (ROS). Mammalian mitochondrial complex I dynamically incorporates 45 distinct subunits into a ~ 1 MDa mature structure (***Vinothkumar et al., 2014***; ***Guerrero-Castillo et al., 2017***). It is known that oxidatively damaged subunits can be exchanged in the intact holo-enzyme (***Dieteren et al., 2012***), but how this process may be regulated is poorly understood. The efficiency and functional integrity of OXPHOS are thought to be partly maintained through a combination of tightly co-ordinated transcriptional and post-transcriptional regulation

**eLife digest** Animal cells generate over 90% of the energy they need within small structures called mitochondria. Converting food into energy requires many different proteins and cells control the relative amounts of the proteins in mitochondria to ensure this process is efficient. To make more of a given protein, the cell must copy the DNA of the gene that encodes it into another molecule known as a messenger RNA, before reading the instructions in the messenger RNA to build the protein. However, this is not the only way that a cell uses molecules of RNA.

A second group of RNAs called long non-coding RNAs (or lncRNAs) can help regulate the production of proteins in complex ways, and each lncRNA can have an effect across multiple genes. Some lncRNAs, for example, stop a third group of RNAs – microRNAs – from blocking certain messenger RNAs from being read. Sirey et al. set out to answer whether a lncRNA might help to co-ordinate the production of the many proteins needed by mitochondria.

In experiments with mouse cells grown in the laboratory, Sirey et al. identified a lncRNA called *Cerox1* that can co-ordinate the levels of at least 12 mitochondrial proteins. A microRNA called miR-488-3p suppresses the production of many of these proteins. By binding to miR-488-3p, *Cerox1* blocks the effects of the microRNA so more proteins are produced. Sirey et al. artificially altered the amount of *Cerox1* in the cells and showed that more *Cerox1* leads to higher mitochondria activity. Further experiments revealed that this same control system also exists in human cells.

Mitochondria are vital to cell survival and changes that affect their efficiency can be fatal or highly debilitating. Reduced efficiency is also a hallmark of ageing and contributes to conditions including cardiovascular disease, diabetes and Parkinson's disease. Understanding how mitochondria are regulated could unlock new treatment methods for these conditions, while a better understanding of the co-ordination of protein production offers other insights into some of the most fundamental biology.

DOI: https://doi.org/10.7554/eLife.45051.002

(*Mootha et al., 2003*; *van Waveren and Moraes, 2008*; *Sirey and Ponting, 2016*) and specific sub-cytoplasmic co-localisation (*Matsumoto et al., 2012*; *Michaud et al., 2014*). The nuclear encoded subunits are imported into the mitochondria after translation in the cytoplasm and their complexes assembled together with the mitochondrially encoded subunits in an intricate assembly process (*Perales-Clemente et al., 2010*; *Lazarou et al., 1793*; *Vogel et al., 2007*). Mitochondrial biogenesis is co-ordinated first transcriptionally from both genomes (*Scarpulla et al., 2012*), and then post-transcriptionally by regulatory small noncoding RNAs such as microRNAs (miRNAs) (*Dumortier et al., 2013*; *Li et al., 2012*). Recently, SAMMSON a long noncoding RNA (lncRNA) was found to bind p32 and, within mitochondria, enhanced the expression of mitochondrial genome-encoded polypeptides (*Leucci et al., 2016*).

Nuclear-encoded and cytosol-located lncRNAs have not yet been implicated in regulating mitochondrial OXPHOS (*Vendramin et al., 2017*) despite being surprisingly numerous and often found localised to mitochondrion- and ribosome-adjacent portions of the rough endoplasmic reticulum (*van Heesch et al., 2014*). It is here, on the ribosome, that turnover of miRNA-targeted mRNAs frequently occurs during their translation (*Tat et al., 2016*). Here we describe a novel mammalian conserved lncRNA, termed *Cerox1* (cytoplasmic endogenous regulator of oxidative phosphorylation 1). *Cerox1* regulates complex I activity by co-ordinately regulating the abundance of at least 12 complex I transcripts via a miRNA-mediated mechanism. *Cerox1* knockdown decreases the enzymatic activities of complexes I and IV. Conversely, elevation of *Cerox1* levels increases their enzymatic activities, halves cellular oxidative stress, and protects cells against the cytotoxic effects of the complex I inhibitor rotenone. To our knowledge, *Cerox1* is the first lncRNA modulator of normal mitochondrial energy metabolism homeostasis and cellular redox state. The miRNA-dependency of *Cerox1* and the regulation of associated OXPHOS transcripts are supported by: (i) direct physical interaction of miR-488–3p with *Cerox1* and complex I transcripts; (ii) decrease or increase in *Cerox1* and complex I transcripts following miR-488–3p overexpression or inhibition, respectively; (iii) miR-488–3p destabilisation of wildtype *Cerox1*, but not a *Cerox1* transcript containing a mutated miR-488–3p miRNA recognition element (MRE) seed region; and, (iv) absence of the OXPHOS

phenotypes either in cell lines deficient in microRNA biogenesis or when *Cerox1*'s predicted miR-488–3p response element is mutated. The miRNA-dependent role of *Cerox1* illustrates how RNA-interaction networks can regulate OXPHOS and that lncRNAs represent novel targets for modulating OXPHOS enzymatic activity.

## Results

### *Cerox1* is a conserved, highly expressed long noncoding RNA

*Cerox1* was selected for further investigation from among a set of central nervous system-derived polyadenylated long non-coding RNAs identified by cDNA sequencing (GenBank Accession AK079380, 2810468N07Rik) (*Carninci et al., 2000*; *Ponjavic et al., 2007*). Mouse *Cerox1* is a 1.2 kb, two exon, intergenic transcript which shares a bidirectional promoter with the SRY (sex determining region Y)-box 8 (*Sox8*) gene (*Figure 1A*). A human orthologous transcript (*CEROX1*, GenBank Accession BC098409) was identified by sequence similarity and conserved synteny (60–70% nucleotide identity within alignable regions, *Figure 1B,C*). Both mouse and human transcripts have low protein coding potential (Materials and methods, *Figure 1—figure supplement 1A*) and no evidence for translation from available proteomic datasets.

Four human or mouse data types supported *CEROX1* as having an important organismal role. First, its promoter shows a greater extent of sequence conservation than the adjacent *SOX8* promoter and its exons are conserved among eutherian mammals (*Figure 1B*). Second, from expression data, its levels in primary tissues and cells are exceptionally high, within the top 13% of a set of 879 lncRNAs with associated cap analysis of gene expression (CAGE) clusters (*Figure 1D,E*). Expression is particularly high in neuroglia, neural progenitor cells and oligodendrocyte progenitors (*Bergmann et al., 2015*; *Mercer et al., 2010*; *Zhang et al., 2014*) (*Figure 1—figure supplement 1B*). *Cerox1* is notable in its higher expression in the adult brain than 64% of all protein coding genes. *Cerox1* expression is also developmentally regulated. For example, it is a known marker gene for type one pre-haematopoietic stem cells in the dorsal aorta (*Zhou et al., 2016*) and its expression is high in mouse embryos beyond the 21 somite stage (https://dmdd.org.uk/). High expression of *Cerox1* in the brain was confirmed using quantitative real-time PCR (qPCR) for both mouse and human orthologous transcripts (*Figure 1—figure supplement 1C,D*).

Third, single nucleotide variants are significantly associated both with *CEROX1* expression and anthropomorphic traits, measured in the UK Biobank. These lie within an 85 kb interval encompassing the *CEROX1* locus, and 5' regions of *SOX8* and *LMF1* genes. For example, rs3809674 is significantly associated with standing and sitting heights; arm fat-free mass (left or right); arm predicted mass (left or right); trunk fat-free or predicted mass; and whole body fat-free mass ($p<5\times10^{-8}$, *Supplementary file 1*); this variant is also a *CEROX1* expression quantitative trait locus (eQTL) for 30 GTEx tissues ($p\leq0.05$), some with absolute normalised effect sizes reaching 0.83 (*Figure 1—figure supplement 1E*). Genetically determined expression change in *CEROX1* therefore could explain, in part, variation in these anthropomorphic traits. These variations affect a large fraction of the human population (minor allele frequency of 28% in 1000 Genomes data). An alternative interpretation that rs3809674 (and linked variants) act on anthropomorphic traits through *LMF1*, rather than *CEROX1*, is consistent with this variant being an eQTL for *LMF1*, but is not consistent with the known protein function of LMF1, a lipase maturation factor, because the associated anthropomorphic traits relate only to fat-free mass. A summary-data-based Mendelian randomization analysis that uses rs3809674 as an instrumental variable, predicts an effect of *CEROX1* gene expression on glioma risk (*Melin et al., 2017*). Finally, it has recently been demonstrated in a mouse model of haematopoietic lineage differentiation that *Cerox1* depletion may impair the contributions of stem cell or B cell differentiation to haematopoiesis (*Delás et al., 2019*).

In mouse neuroblastoma (N2A) cells the most highly expressed *Cerox1* transcript (*Figure 1—figure supplement 1F,G*) is enriched in the cytoplasmic fraction (*Figure 1F*) with a short half-life of 36 ± 16 mins (*Figure 1—figure supplement 1H*) and is mainly associated with the ribosome-free fraction (*Figure 1—figure supplement 1I*).

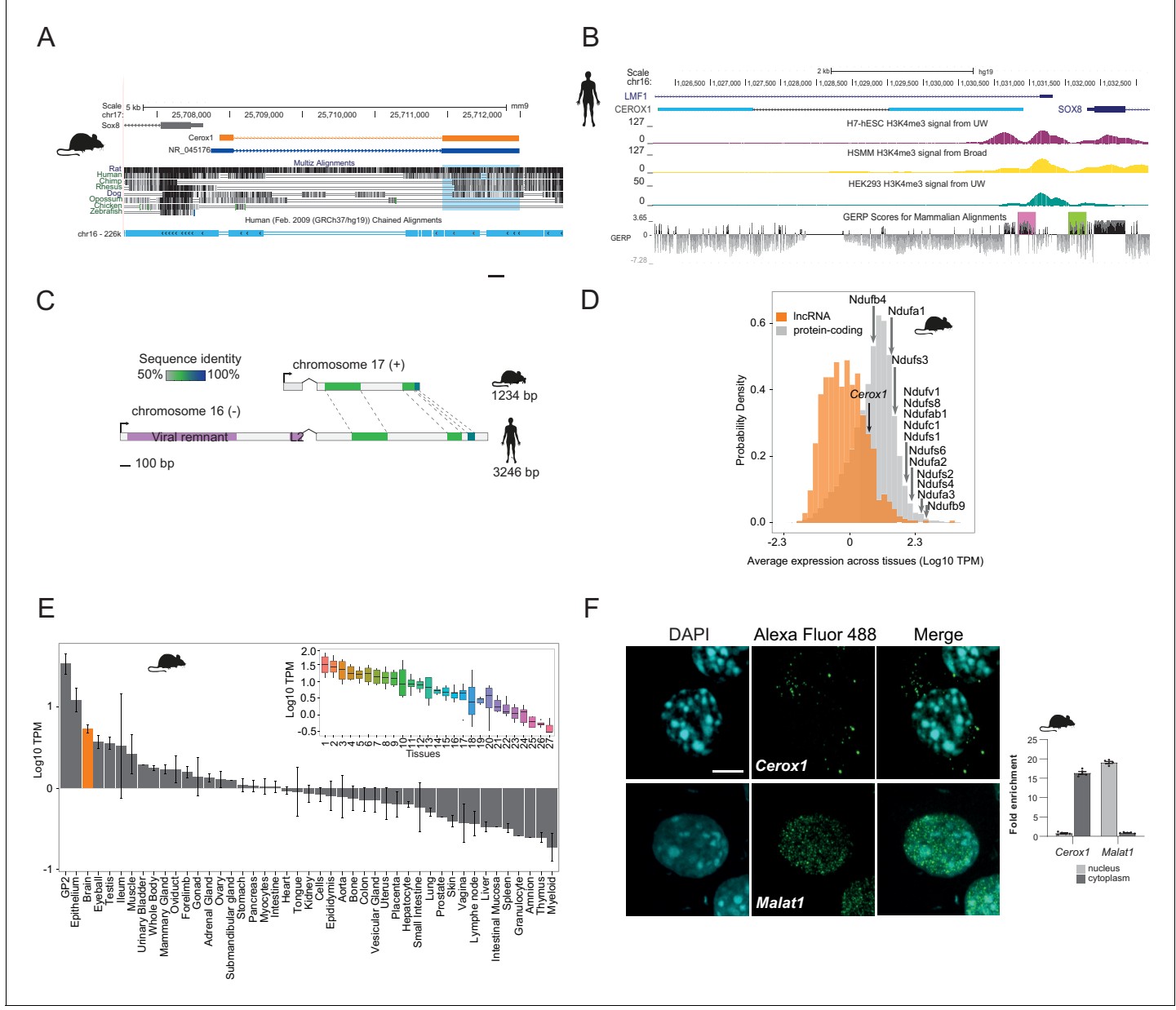

**Figure 1.** *Cerox1* is an evolutionarily conserved, highly expressed and predominantly cytoplasmic lncRNA. (**A**) The mouse *Cerox1* locus (mm9 assembly). Sequence shaded in blue highlights conservation within exon two among eutherian mammals, but not in non-mammalian vertebrates such as chicken and zebrafish. (**B**) Syntenic human locus (hg19). This transcript was previously identified on the minus strand as *LMF1* non-coding transcript 4, and is located within the intron of *LMF1* non-coding transcript 2. *LMF1* non-coding transcript two is annotated as a *nonsense mediated decay* biotype. A prominent peak of H3K4me3 (tri-methylation of histone H3 lysine 4 often found near promoters) modification marks the *CEROX1* transcriptional start site. H3K4me3 peaks for H7-human embryonic stem cells (H7-hESC), human skeletal muscle myoblasts (HSMM) and human embryonic kidney 293 cells (HEK293) are depicted. The genomic evolutionary rate profiling (GERP) score indicates a higher extent of conservation of the *CEROX1* promoter (shaded pink) than the adjacent *SOX8* promoter (shaded green). (**C**) Schematic representation of mouse *Cerox1* transcript and the human orthologous sequence. Exon two contains blocks of 60–70% sequence identity; human *CEROX1* has an additional 1235 bases of retrotransposed insertions at the 5' end. (**D**) Distributions of lncRNAs' and protein-coding genes' average expression levels across tissues in mouse. Average expression levels of representative mitochondrial complex I subunits' mRNAs are indicated. TPM = tags per million. (**E**) Average expression levels of *Cerox1* across mouse tissue samples. The orange bar highlights nervous system tissue samples whose values for replicates among neurological tissues are shown in the inset panel: 1- Medulla oblongata, 2– Spinal cord, 3– Diencephalon, 4– Substantia nigra, 5– Microglia, 6– Raphe, 7– Dorsal spinal cord, 8– Corpora quadrigemina, 9– Cortex, 10– Corpus striatum, 11- Visual cortex, 12– Olfactory brain, 13– Cerebellum, 14– Neurospheres sympathetic neuron derived, 15– Neurospheres parasympathetic neuron derived, 16– Neurospheres enteric neuron derived, 17– Astrocytes (cerebellar), 18– Hippocampus, 19– Hippocampal, 20– Ventral spinal cord, 21– Astrocytes, 22– Pituitary gland, 23– Astrocytes (hippocampus), 24– Cortical neurons, 25– Striatal neurons, 26– Schwann cells, 27– Meningeal cells. Error bars indicate s.e.m. (**F**) Cytoplasmic localisation of mouse *Cerox1* compared to a nuclear retained lncRNA,

*Figure 1 continued on next page*

*Figure 1 continued*

*Malat1*, as demonstrated by fluorescent in situ hybridization and cell fractionation followed by quantitative PCR. By fractionation, mouse *Cerox1* is 15-fold enriched in the cytoplasm of N2A cells (*n* = 5; error bars s.e.m.). Scale bar = 5 μm.

DOI: https://doi.org/10.7554/eLife.45051.003

The following figure supplement is available for figure 1:

**Figure supplement 1.** Transcript characterisation.

DOI: https://doi.org/10.7554/eLife.45051.004

## *Cerox1* expression modulates levels of oxidative phosphorylation transcripts

Expression of *Cerox1* was manipulated by transient overexpression or shRNA-mediated knockdown. Twelve shRNAs were tested for the ability to knock-down *Cerox1* (*Figure 2—figure supplement 1A*). One of these (sh92) decreased expression levels by greater than 60%, with the next best shRNA (sh1159) decreasing expression by approximately 40%. As expected from their sharing of the *Cerox1-Sox8* bidirectional promoter, CRISPR-mediated activation and inhibition of the *Cerox1* locus was not specific as it led also to changes in *Sox8* expression (*Figure 1A*, *Figure 2—figure supplement 1B*). However, decreasing *Cerox1* levels in N2A cells by shRNA or transient overexpression had no effect on the expression of neighbouring genes (*Figure 2—figure supplement 1C*). In contrast, *Cerox1* overexpression led to differential expression of 286 distal genes (*q* < 0.05, Bonferroni multiple testing correction; *Supplementary file 2*), of which an unexpected and large majority (83%; 237) were upregulated (p<$10^{-6}$; binomial test). Our attention was immediately drawn to the considerable (≥20 fold) enrichment of the mitochondrial respiratory chain gene ontology term among upregulated genes (*Figure 2A*).

The mitochondrial electron transport chain (ETC) consists of five multi-subunit complexes encoded by approximately 100 genes of which only 13 are located in the mitochondrial genome. The 15 ETC transcripts that show statistically significant differential expression after *Cerox1* overexpression are nuclear encoded (*Figure 2B,C*) with the greatest changes observed by qPCR for complex I subunit transcripts (*Figure 2—figure supplement 1D*). Twelve of 35 nuclear encoded complex I subunits or assembly factors transcripts increased substantially and significantly (>40%) following *Cerox1* overexpression; we consider these to be gene expression biomarkers for *Cerox1* activity in the mouse N2A system (*Figure 2C*). In the reciprocal *Cerox1* knock-down experiment, all 12 were reduced in abundance using sh92, three significantly, with a concordant pattern observed for the less effective shRNA, sh1159 (*Figure 2—figure supplement 1E,F*). Taken together, these results indicate that *Cerox1* positively and co-ordinately regulates the levels of many mitochondrial complex I transcripts.

Increased abundance of OXPHOS subunit transcripts, following *Cerox1* overexpression, was found to elevate protein levels. Western blots using reliable antibodies for the key complex I catalytic core proteins NDUFS1 and NDUFS3 showed approximately 2.0-fold protein level increases that surpassed their ~ 1.4 fold transcript level changes (median 2.4 and 1.4-fold increases [p=0.0013 and 0.002], respectively; *Figure 2D*). *Cerox1* transcript abundance is thus coupled positively to OXPHOS transcript levels and to their availability for translation, resulting in an amplification of the amount of protein produced. In summary, protein subunits of the same complex (Complex I) that are sustained at high abundance and with long half-lives (*Dörrbaum et al., 2018*; *Mathieson et al., 2018*; *Schwanhäusser et al., 2011*) (*Figure 2—figure supplement 1G*), and whose transcripts are stable (*Schwanhäusser et al., 2011*; *Friedel et al., 2009*; *Sharova et al., 2009*; *Tani et al., 2012*) (*Figure 2—figure supplement 1H*) and also have very high copy numbers in the cell (*Schwanhäusser et al., 2011*; *Cao et al., 2017*), can be increased two-fold, and co-ordinately, by the simple expediency of increasing the level of this single abundant lncRNA.

These large effects on protein and mRNA copy number are associated with both metabolic and cellular phenotypes. Ten metabolites are significantly different in N2A cells overexpressing *Cerox1* (*Figure 2E*). These cells show a significant increase in the reduced glutathione to oxidized glutathione ratio (GSH:GSSG, p=1.3×$10^{-3}$, *Figure 2D* Inset) indicative of a more favourable cellular redox state. *Cerox1*-overexpressing cells also exhibited a 43% reduction in cell cycle activity, yet without a

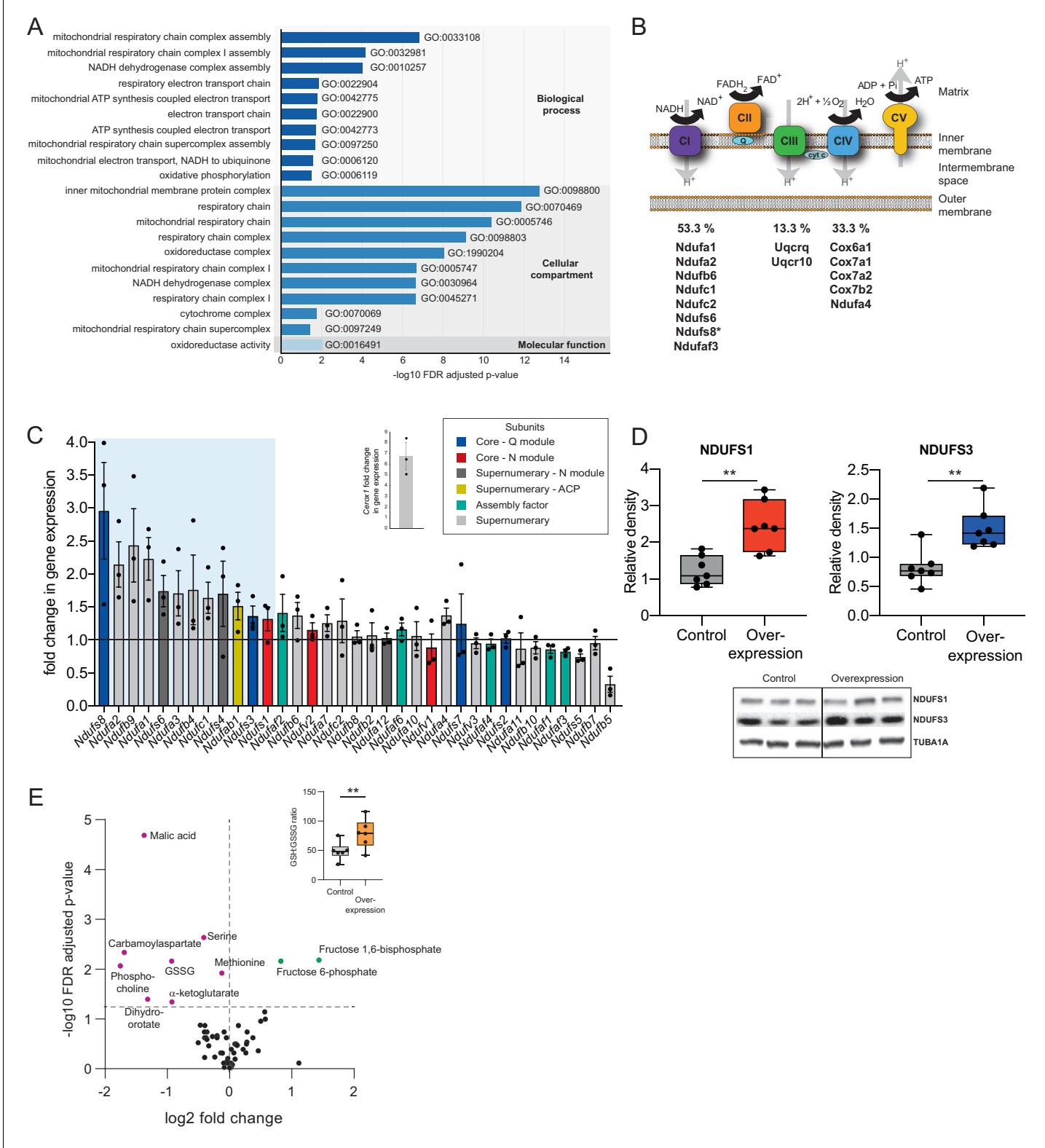

**Figure 2.** *Cerox1* overexpression elevates levels of OXPHOS transcripts and their encoded proteins. (**A**) Gene ontology analysis indicates a significant enrichment of upregulated genes involved in mitochondrial electron transport, energy production and redox reactions. (**B**) Four membrane bound multi-subunit complexes (CI, CII, CIII, CIV) are embedded in the inner mitochondrial membrane and facilitate transfer of electrons; three of these subunits are also proton pumps which create the chemiosmotic gradient required for ATP synthase activity, with complex V being ATP synthase. The subunits vary in size and complexity with Complex I (NADH:ubiquinone oxidoreductase) consisting of 45 subunits, Complex II (succinate

*Figure 2 continued on next page*

*Figure 2 continued*

dehydrogenase) four subunits, Complex III (Ubiquinol:cytochrome *c* oxidoreductase) 11 subunits and Complex IV (Cytochrome *c* oxidase) 13 subunits. Of 15 oxidative phosphorylation genes whose transcripts were up-regulated following *Cerox1* overexpression 53% were subunits of Complex I, 13% were subunits of Complex III and 33% were subunits of Complex IV. * indicates core subunits that are essential for activity. Note: subunit NDUFA4 has recently been reassigned to mitochondrial complex IV (*Balsa et al., 2012*). (C) qPCR profiling of 35 complex I subunits and assembly factors (30 nuclear encoded complex I subunits and five assembly factors). Transcripts showing a 1.4 fold, or greater, change in expression after overexpression of *Cerox1* are present within the boxed shaded area. Fold change of wild-type *Cerox1* compared to the control are indicated in the inset panel. The transcripts profiled can be characterised into six categories: Core–Q module, subunits responsible for the electron transfer to ubiquinone; Core–N module, subunits responsible for the oxidation of NADH; Supernumerary subunits– those that are additional to the core subunits required for the catalytic role of complex I, but do not play a catalytic role themselves. Many of these subunits may be performing a structural role, but the majority are of unknown function. The supernumerary subunits can be further subdivided into supernumerary – N module, those accessory subunits associated with the NADH oxidation module of CI; supernumerary ACP (acyl carrier protein) – in addition to being a non-catalytic subunit of CI, NDUFAB1 is also a carrier of the growing fatty acid chain in mitochondrial fatty acid biosynthesis; assembly factor, proteins that are required for the correct assembly and integration of CI. Error bars s.e.m. (*n* = 3 biological replicates). (D) Overexpression of *Cerox1* results in large increases in the total protein levels of two core subunits for which high quality antibodies exist, normalised to the loading control α-tubulin (TUBA1A). NDUFS1 is one of three (NDUFS1, NDUFV1, NDUFV2) core components of the N-module of Complex I. NDUFS3 is one of four (NDUFS2, NDUFS3, NDUFS7, NDUFS8) core components of the Complex I Q-module. *n* = 7 biological replicates for control and overexpression. 2-sided *t*-test; **p<0.01. (E) Overexpression of *Cerox1* results in a change in the metabolite profile of N2A cells, with 10 of 66 metabolites measured demonstrating a significant change in the experimental sample after multiple testing correction (q < 0.05; *n* = 6 biological replicates for pCAG-control and pCAG *Cerox1* overexpression). N2A cells overexpressing *Cerox1* show an increased GSH:GSSG ratio (figure inset, 2-sided *t*-test; **p<0.01).

DOI: https://doi.org/10.7554/eLife.45051.005

The following source data and figure supplement are available for figure 2:

**Source data 1.** N2A metabolomics profiling - *Figure 2E*.
DOI: https://doi.org/10.7554/eLife.45051.007
**Figure supplement 1.** Manipulation of *Cerox1* expression levels.
DOI: https://doi.org/10.7554/eLife.45051.006

change in the proportion of live/dead cells or a deviation from normal cell cycle proportions (*Figure 2—figure supplement 1I,J,K*). *Cerox1* levels thus affect the overall timing of cell division.

## *Cerox1* can regulate mitochondrial OXPHOS enzymatic activity

Increased translation of some complex I transcripts leads to increased respiration (*Shyh-Chang et al., 2013*) and, more specifically, to an increase in the enzymatic activity of complex I (*Alvarez-Fischer et al., 2011*). To address this hypothesis we used oxidative phosphorylation enzyme assays to investigate whether changes in expression to a subset of subunits lead to a change in enzyme activity and oxygen consumption. Indeed, complex I and complex IV enzymatic activities increased substantially after *Cerox1* overexpression (by 22%, p=0.01; by 50%, p=0.003, respectively; 2-tailed Student's *t*-test; *Figure 3A*). Such rate increases for two of the eukaryotic cell's most abundant and active enzymes were unexpected.

We next measured oxygen consumption under these conditions using a Seahorse XF$^e$24 Analyzer. These complexes' more rapid catalytic rates resulted, unexpectedly, in large increases in: (i) overall basal oxygen consumption (by 85%), (ii) ATP-linked oxygen consumption (by 107%) and, (iii) maximum uncoupled respiration (by 59%; p=$5 \times 10^{-4}$, p=$1 \times 10^{-4}$, p=$4 \times 10^{-3}$, respectively, 2-tailed Student's *t*-test; *Figure 3B*). These increases in enzyme activities and mitochondrial respiration are expected to produce persistent and substantial increases beyond the already very high basal rate of ATP formation (*Rich, 2003*), due to the long-half lives of the *Cerox1* sensitive complex I protein subunits (*Dörrbaum et al., 2018*; *Mathieson et al., 2018*; *Schwanhäusser et al., 2011*).

Conversely, after sh92-mediated *Cerox1* knockdown complex I and complex IV enzymatic activities decreased significantly (by 11%, p=0.03% and 19%, p=0.02, respectively; *Figure 3C*), with concomitant large decreases in basal oxygen consumption (53%), ATP linked oxygen consumption (52%) and maximal uncoupled respiration (61%; p=0.011, p=0.034, p=0.042 respectively, 2-tailed Student's *t*-test; *Figure 3D*).

These observed changes in enzymatic activity were not due to changes in mitochondria number because the enzymatic activities of complexes II, III and citrate synthase (*Figure 3A,B*), and the mitochondrial-to-nuclear genome ratio (*Figure 3—figure supplement 1A,B*), each remained unaltered

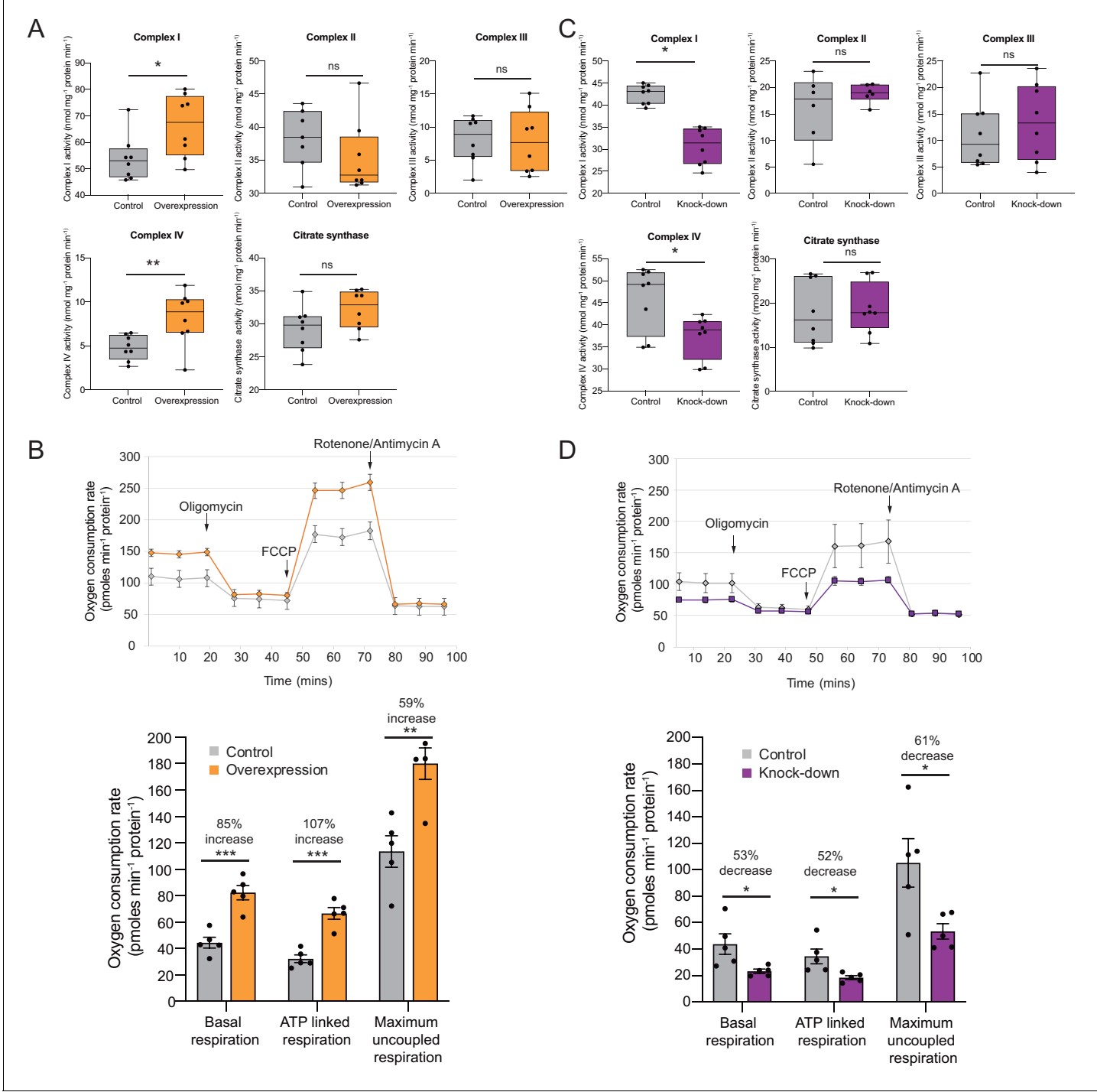

**Figure 3.** OXPHOS enzyme activity and oxygen consumption change concordantly and substantially with *Cerox1* level alteration. (**A**) Enzyme activities in mouse N2A cells 72 hr post-transfection of *Cerox1* overexpression construct. Mouse *Cerox1* overexpression in N2A cells results in significant increases in the catalytic activities of complexes I (22% increase) and IV (50% increase). Complexes II, III and citrate synthase show no significant change in activity. *n* = 8 biological replicates for control and overexpression. (**B**) Oxygen consumption, by N2A cells overexpressing *Cerox1*. Top: normalised real time oxygen consumption rate in basal conditions and after sequential injections of oligomycin, FCCP and rotenone/antimycin A. Bottom: changes in basal, ATP-linked and maximum uncoupled respiration respectively. Error bars s.e.m. (*n* = 5 biological replicates). (**C**) sh92-mediated knockdown of *Cerox1* results in significant decreases of Complexes I and IV enzymatic activities 72 hr post transfection; no significant changes were observed for complexes II, III or the citrate synthase control. *n* = 8 biological replicates for control and knockdown. The less effective shRNA (sh1159) also decreased oxygen consumption yet not significantly relative to the control (data not shown). (**D**) Oxygen consumption, by *Cerox1* knockdown N2A cells. Top: normalised real time oxygen consumption rate in basal conditions and after sequential injections of oligomycin, FCCP and rotenone/antimycin A.

*Figure 3 continued on next page*

*Figure 3 continued*

Bottom: changes in basal, ATP-linked and maximum uncoupled respiration respectively. Error bars s.e.m. ($n$ = 5 biological replicates). 2-tailed Student's *t*-test: ***p<0.001, **p<0.01, *p<0.05, ns not significant.

DOI: https://doi.org/10.7554/eLife.45051.008

The following source data and figure supplement are available for figure 3:

**Source data 1.** N2A Cerox1 overexpression specific enzyme assays - *Figure 3A*.
DOI: https://doi.org/10.7554/eLife.45051.010
**Source data 2.** N2A Cerox1 overexpression seahorse bioanalyzer - *Figure 3B*.
DOI: https://doi.org/10.7554/eLife.45051.011
**Source data 3.** N2A Cerox1 knock down specific enzyme assays - *Figure 3C*.
DOI: https://doi.org/10.7554/eLife.45051.012
**Source data 4.** N2A Cerox1 knock down seahorse bioanalyzer - *Figure 3D*.
DOI: https://doi.org/10.7554/eLife.45051.013
**Figure supplement 1.** Increases in mitochondrial complex I and complex IV activities are not due to an increase in mitochondrial copy number.
DOI: https://doi.org/10.7554/eLife.45051.009

by changes in *Cerox1* levels. These data indicate that *Cerox1* can specifically and substantially regulate oxygen consumption and catalytic activities of complex I and complex IV in mouse N2A cells.

## *Cerox1* expression can protect cells from oxidative stress

Complex I deficient patient cells experience elevated ROS production (*Pitkanen and Robinson, 1996*). In *Cerox1* knockdown N2A cells ROS levels were increased significantly, by almost 20% (p=$4.2\times10^{-6}$; *Figure 4A*). Conversely, in cells overexpressing *Cerox1*, ROS production was nearly halved (p=$3.5\times10^{-7}$; *Figure 4A*), and protein carbonylation, a measure of ROS-induced damage, was reduced by 35% (p=$1\times10^{-3}$; *Figure 4B*). Knock-down of *Cerox1* resulted in a 6.6% increase in protein carbonylation compared to the control (p=0.05, data not shown). The observed *Cerox1*-dependent reduction in ROS levels is of particular interest because mitochondrial complex I is a major producer of ROS which triggers cellular oxidative stress and damage, and an increase in ROS production is a common feature of mitochondrial dysfunction in disease (*Murphy, 2009*).

We next demonstrated that the increased activities of complex I and complex IV induced by *Cerox1* protect cells against the deleterious effects of specific mitochondrial complex inhibitors, specifically rotenone and sodium azide (complex I and complex IV inhibitors, 37% and 58% respectively, p<0.01); conversely, *Cerox1*-knockdown cells were significantly more sensitive to rotenone and exposure to heat (12%, p<0.001% and 11%, p<0.01 respectively *Figure 4C*). Cells overexpressing *Cerox1* and treated with rotenone, a complex I inhibitor, exhibited no significant difference in protein carbonylation (data not shown). Taken together, these results indicate that elevation of *Cerox1* expression leads to decreased ROS production, decreased levels of oxidative damage to proteins and can confer protective effects against complex I and complex IV inhibitors.

## Increased OXPHOS enzymatic activity is dependent upon miRNA binding to *Cerox1*

Due to their positive correlation in expression and cytoplasmic localisation we next considered whether *Cerox1* regulates complex I transcripts post-transcriptionally by competing with them for the binding of particular miRNAs. To address this hypothesis, we took advantage of mouse Dicer-deficient ($Dicer^{\Delta/\Delta}$) embryonic stem cells that are deficient in miRNA biogenesis (*Nesterova et al., 2008*). We first tested *Cerox1* overexpression in wildtype mouse ES cells and showed that this, again, led to an increase in transcript levels, specifically of six complex I subunits (*Figure 5A*), of which four had previously shown significant changes in N2A cells after *Cerox1* overexpression (*Figure 2C*). In contrast, overexpression in $Dicer^{\Delta/\Delta}$ cells failed to increase levels of these transcripts (*Figure 5A*). These results indicate that *Cerox1*'s ability to alter mitochondrial metabolism is miRNA-dependent.

Four miRNA families (miR-138–5p, miR-28/28–5p/708–5p, miR-370–3p, and miR-488–3p) were selected for further investigation based on the conservation of their predicted binding sites (MREs) in both mouse *Cerox1* and human *CEROX1* (*Figure 5B*). All five MREs conserved in mouse *Cerox1* and human *CEROX1* for N2A-expressed miRNAs (*Figure 5B Figure 5—figure supplement 1A*)

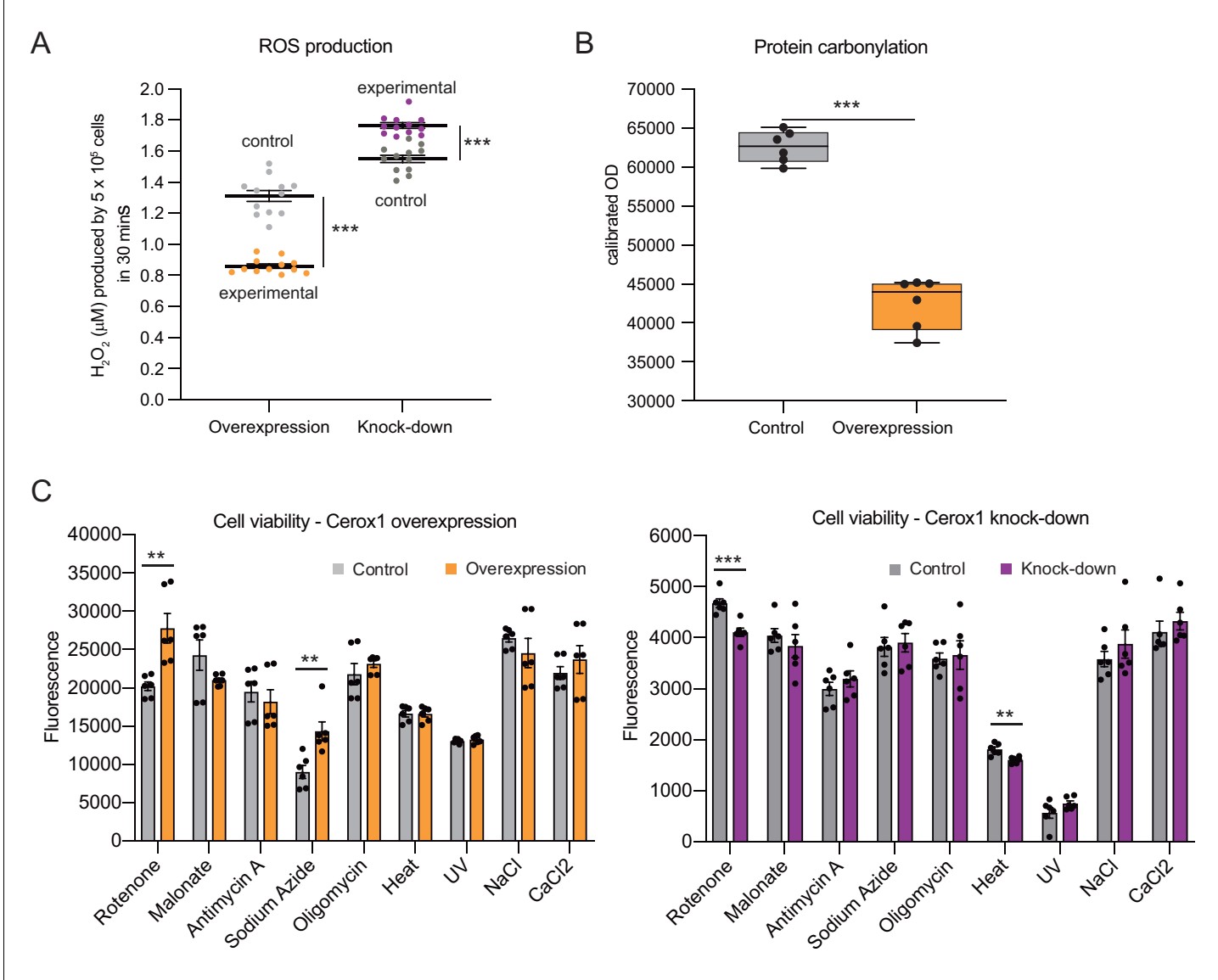

**Figure 4.** Cellular oxidative stress and viability depend on *Cerox1* levels. (**A**) *Cerox1* knockdown increases the production of reactive oxygen species by 20%, whilst *Cerox1* overexpression decreases it by 45% (error bars s.e.m., *n* = 12 biological replicates). (**B**) Protein oxidative damage also decreases in the overexpression condition compared to the control, as measured by densitometry on western blots against carbonylation of amino acid side chains. *n* = 6 biological replicates. (**C**) Viability of *Cerox1* overexpressing and knock-down cells when stressed. N2A cells were stressed by addition of electron transport chain (ETC) inhibitors (rotenone, CI inhibitor; malonate, competitive inhibitor of CII; antimycin A, CIII inhibitor; sodium azide, CIV inhibitor; oligomycin, ATP synthase inhibitor), exposure to environmental stress (heat, ultraviolet radiation), or manipulation of extracellular osmolarity (NaCl) or extracellular calcium (CaCl₂) concentration, for 1 hr and then the viability of the cells measured using the fluorescent indicator Alamar Blue. Error bars s. e.m. (*n* = 6 biological replicates for overexpression control, overexpression, knock-down control and knock-down). 2-tailed Student's *t*-test: ***p<0.001, **p<0.01.

DOI: https://doi.org/10.7554/eLife.45051.014

The following source data is available for figure 4:

**Source data 1.** N2A Reactive oxygen species production - *Figure 4A*.
DOI: https://doi.org/10.7554/eLife.45051.015
**Source data 2.** N2A Cerox1 overexpression protein carbonylation - *Figure 4B*.
DOI: https://doi.org/10.7554/eLife.45051.016
**Source data 3.** N2A cell viability Cerox1 overexpression and knockdown - *Figure 4C*.
DOI: https://doi.org/10.7554/eLife.45051.017

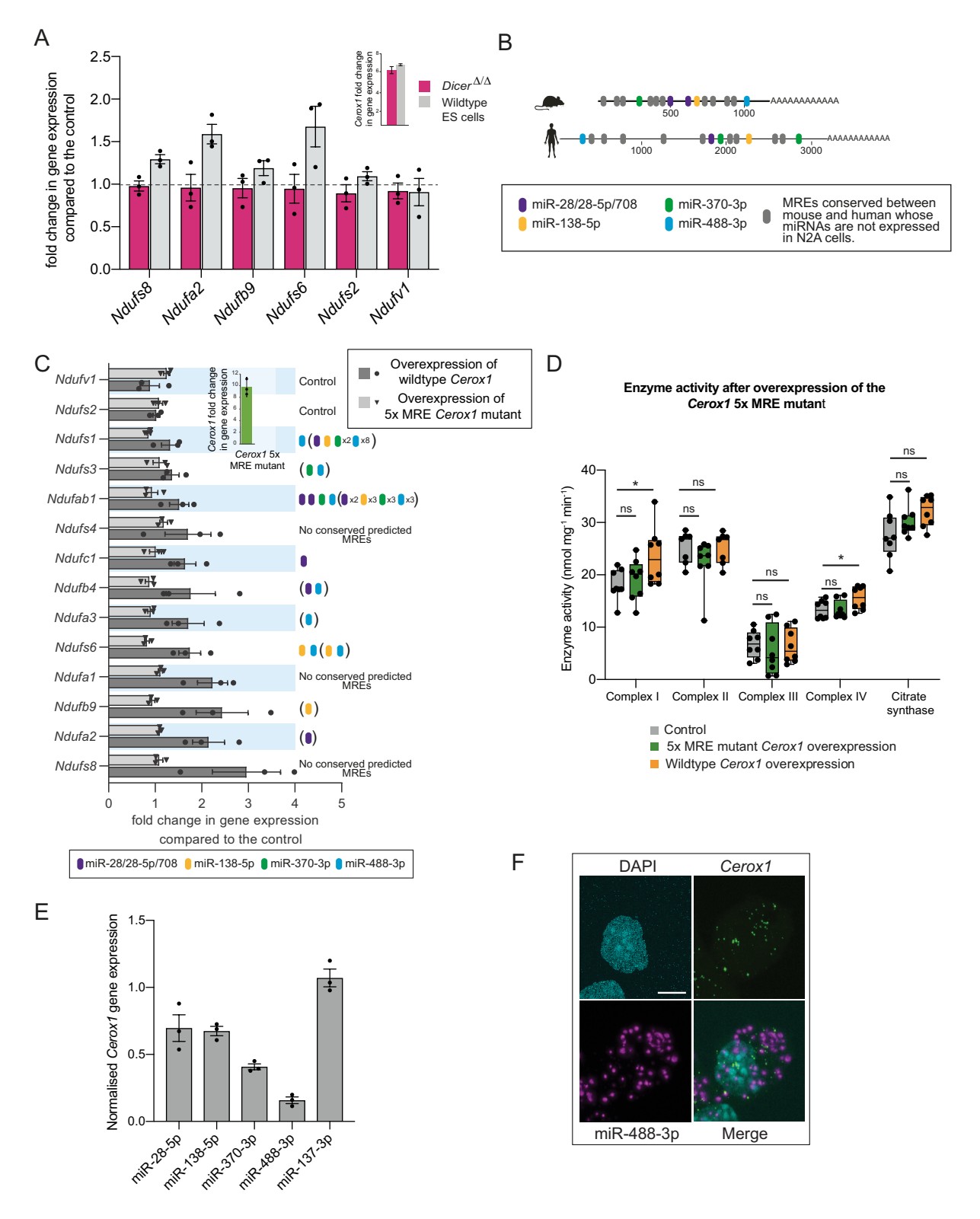

**Figure 5.** The effect of *Cerox1* on complex I transcript levels is miRNA-dependent. (**A**) Overexpression of *Cerox1* in mouse wildtype and *Dicer*^Δ/Δ embryonic stem (ES) cells (inset graph). The overexpression of *Cerox1* in wildtype mouse embryonic stem cells results in an increase in complex I subunit transcripts, with no observed change in expression of two control subunits (*Ndufs2, Ndufv1*) that were also unaffected in N2A cells. Overexpression of *Cerox1* in *Dicer*^Δ/Δ embryonic stem cells results in no increase in the expression of any complex I subunit. 2-sided *t*-test; **p<0.01,

*Figure 5 continued on next page*

*Figure 5 continued*

*p<0.05, ns = not significant. Error bars s.e.m. *n* = 3 biological replicates. (**B**) Predicted MREs whose presence is conserved in both the mouse and human *Cerox1*. Coloured MREs indicate those MREs whose presence is conserved between mouse and human and whose miRNAs are expressed in N2A cells. miRNA site types are as follows: miR-28–5p, 8mer-A1; miR-138–5p, 6mer; miR-370–3p, 7mer-m8; miR-488–3p, 7mer-m8; miR-708–5p, 7mer-A1. The grey predicted MREs represent those that are conserved, but whose miRNAs are not expressed in N2A cells (miR-125a-3p, miR-199/199–5p, miR-302ac/520f, miR-485/485–5p, miR-486/486–5p, miR-501/501–5p, miR-654–3p, miR-675/675–5p). (**C**) Overexpression of the 5xMRE mutant failed to alter expression of complex I subunit transcripts that otherwise all increase in abundance following wild-type *Cerox1* overexpression. Fold changes of wildtype *Cerox1* or the 5xMRE *Cerox1* mutant compared to the control are indicated in the inset panel. The numbers of MREs predicted by TargetScan v7.0 (*Agarwal et al., 2015*) in these transcripts' 3'UTRs for the four conserved, N2A expressed miRNA families are indicated (see also *Supplementary file 3*). Due to known widespread noncanonical miRNA binding (*Helwak et al., 2013*), predictions were also extended across the gene body (bracketed MREs). 2-sided *t*-test; **p<0.01, *p<0.05, ns = not significant. Error bars s.e.m. *n* = 3 biological replicates. (**D**) Overexpression of the 5xMRE mutant failed to alter OXPHOS enzymatic activity compared to the control for any of the complexes measured. A one-way ANOVA was applied to test for differences in activities of the mitochondrial complexes between a control and overexpression of wildtype *Cerox1* and the 5xMRE mutant. A post-hoc Dunnett's test indicated that the overexpression of wildtype *Cerox1* resulted, as expected, in significantly increased complex I and IV activities of 30% and 17% respectively ($F$ [2, 21]=4.9, p=0.017; $F$[2, 20]=4.6, p=0.033), while comparisons for the 5xMRE mutant with the control were not significant. There was no significant difference in the activities of complex II ($F$[2,19]=3.5, p=0.26), complex III ($F$[2,19]=0.08, p=0.5) or citrate synthase ($F$ [2,20]=2.6, p=0.42). *n* = 6 biological replicates. Significance levels, one-way ANOVA, Dunnett's post hoc test *p<0.05. (**E**) Four to six fold overexpression of each of four miRNAs with predicted MREs whose presence is conserved in both mouse and human *Cerox1* resulted in a decrease in *Cerox1* transcript level, with overexpression of miR-488–3p resulting in >90% knock down of *Cerox1*. This was not observed when the miRNA miR-137–3p, which has no predicted MREs in *Cerox1*, was similarly overexpressed. Error bars s.e.m. *n* = 3 biological replicates. (**F**) Fluorescent in situ hybridisation detection of miR-488–3p (magenta) and *Cerox1* (green) in N2A cells. Scale bar = 5 μm. A no probe control (*Figure 5—figure supplement 1D*) indicated some background Fast Red signal (miRNA detection) localised to the nucleus, but no background for Alexa Fluor-488 (lncRNA detection).

DOI: https://doi.org/10.7554/eLife.45051.018

The following source data and figure supplement are available for figure 5:

**Source data 1.** N2A wildtype and MRE mutant Cerox1 overexpression specific enzyme assays - *Figure 5D*.

DOI: https://doi.org/10.7554/eLife.45051.020

**Figure supplement 1.** Expression profile of mmu-miR-488-3p.

DOI: https://doi.org/10.7554/eLife.45051.019

were mutated by inversion of their seed regions. This mutated *Cerox1* transcript failed to alter either complex I transcript levels or enzyme activities when overexpressed in mouse N2A cells (*Figure 5C, D*). This indicates that *Cerox1*'s molecular effects are mediated by one or more of these MREs.

If so, then the overexpression of these miRNAs in turn would be expected to deplete *Cerox1* RNA levels. Indeed, overexpression of each miRNA reduced *Cerox1* levels (*Figure 5E*). Overexpression of the tissue-restricted miRNA miR-488–3p (*Figure 5—figure supplement 1B,C*; *Landgraf et al., 2007*; *Isakova and Quake, 2018*) caused the greatest depletion of the *Cerox1* transcript (*Figure 5E*) indicating that this MRE is likely to be physiologically relevant. Dual fluorescent RNA in situ hybridization of miR-488–3p and *Cerox1* indicates the proximity of these non-coding RNAs within the N2A cell (*Figure 5F*) and that both *Cerox1* and miR-488–3p are localised in the cytoplasm (*Figures 1F* and *5F*). *CEROX1* transcripts are predominantly (94%) localised to ribosomes (*van Heesch et al., 2014*), as is the destabilisation by microRNAs of mRNAs as they are being translated (*Tat et al., 2016*). Together, these observations imply that *Cerox1* and mitochondrial protein mRNAs are targets of miR-488–3p on ribosomes within the rough ER that forms a network around mitochondria (*Wu et al., 2017*; *Eskelinen, 2008*).

## *Cerox1* activity is mediated by miR-488–3p

Our previous results showed that *Cerox1* abundance modulates complex I activity and transcripts (*Figures 2–4*) and that miR-488–3p has the greatest effect in decreasing *Cerox1* transcript levels (*Figure 5E*). To determine whether miR-488–3p modulates complex I transcript levels we overexpressed and inhibited miR-488–3p in N2A cells (*Figure 6A,B*). Results showed that miR-488–3p modulates these transcripts' levels, with overexpression leading to a significant downregulation of all 12 *Cerox1*-sensitive complex I transcripts (*Figure 6A*), whilst, conversely, miR-488–3p inhibition leads to increased expression for 10 of 12 transcripts, of which 4 (*Ndufa2, Ndufb9, Ndufs4* and *Ndufs1*) were significantly increased (*Figure 6B*).

To determine whether the single predicted miR-488–3p MRE in *Cerox1* is required to exert its effects on complex I we created a *Cerox1* transcript containing three mutated nucleotides within this

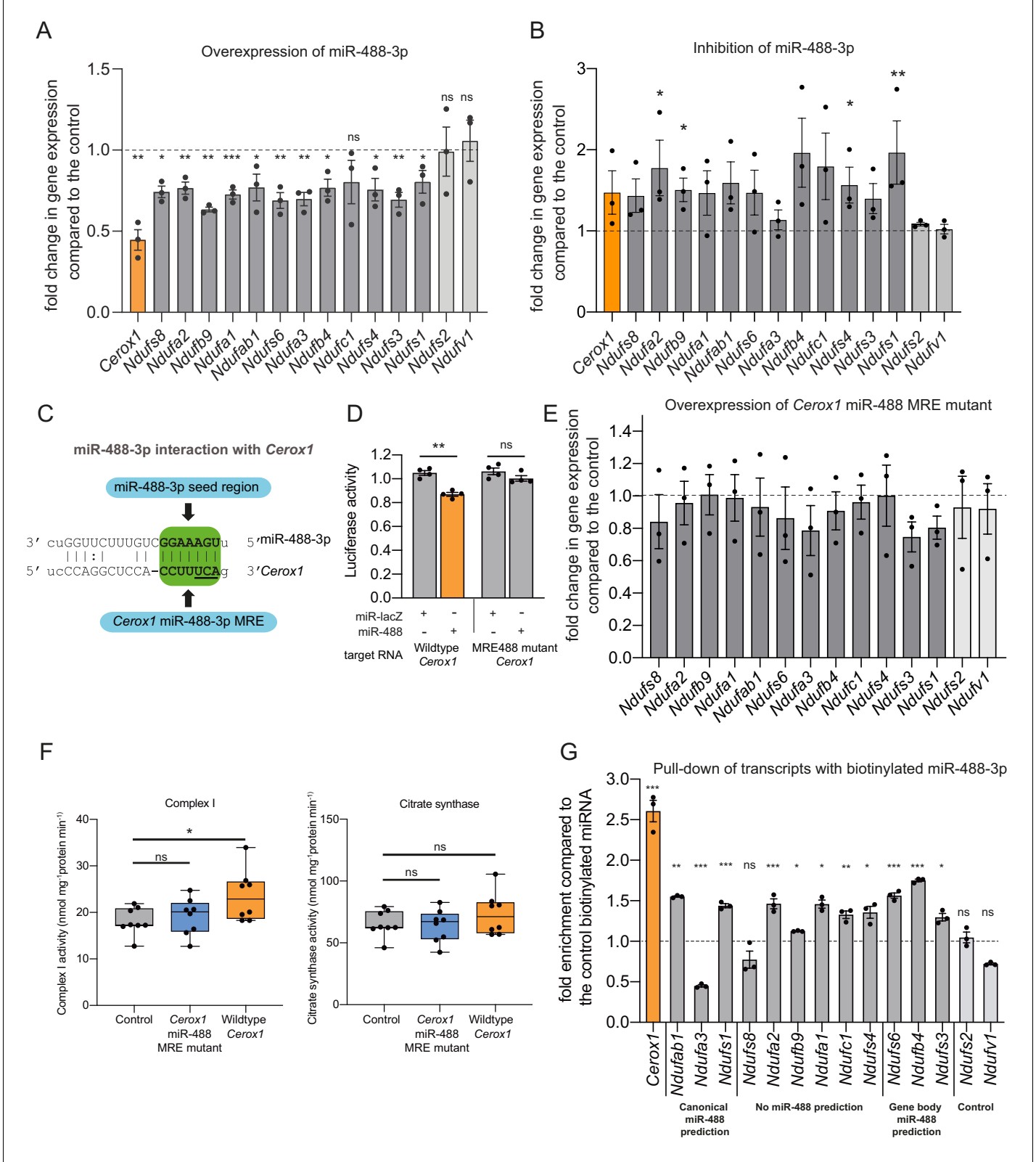

**Figure 6.** An intact miR-488–3p response element site is required for the effect of *Cerox1* on complex I catalytic activity. (**A**) Overexpression of miR-488–3p knocks down all *Cerox1*-sensitive subunit transcripts. Error bars s.e.m. (*n* = 3 biological replicates for control and overexpression of miR-488–3p). 2-sided *t*-test; ***p<0.001, **p<0.01, *p<0.05 (**B**) Inhibition of miR-488–3p increases the expression of most (10/12) target transcripts compared to the control. Error bars s.e.m. (*n* = 3 biological replicates for control and inhibition of miR-488–3p). 2-sided *t*-test; **p<0.01, *p<0.05. (**C**) Schematic of the

*Figure 6 continued on next page*

*Figure 6 continued*
predicted miR-488–3p miRNA recognition element in *Cerox1*. The interaction of miR-488–3p with *Cerox1* is predicted to involve a 7mer-8m seed site, with the heptamer sequence of the seed being complementary to nucleotides 2–8 of the miRNA. Underlined residues indicate the location of the *Cerox1* miR-488-3p MRE mutation. (D) Luciferase destabilisation assay for both wildtype *Cerox1* and *Cerox1* mutated within the miR-488–3p MRE. Error bars s.e.m. (*n* = 4 biological replicates for each condition). 2-sided *t*-test; **p<0.01. (E, F) Overexpression of *Cerox1* mutated within a single miR-488–3p MRE (E) failed to alter expression levels of complex I subunits that increase in expression with wild-type *Cerox1* overexpression (error bars s.e.m, *n* = 3 biological replicates) and (F) failed to recapitulate the increase in complex I catalytic activity observed for the wildtype transcript. Fold change of wildtype *Cerox1* compared to the control is indicated on the left of panel E. As expected, wildtype enzymatic activity was significantly different for complex I (*F* [2, 21]=4.944 *P*=0.019). A post-hoc Dunnett's test indicated that the overexpression of wildtype *Cerox1* resulted in significantly increased complex I activity, while the comparison of the *Cerox1* miR-488–3p MRE mutant with the control was not significant. There was no significant difference in the activity of citrate synthase (*F*[2,21]=1.4, p=0.28). *n* = 8 biological replicates. Significance levels, one-way ANOVA, Dunnett's post hoc test *p<0.05, ns = not significant. (G) Enrichment of 8 *Cerox*1 sensitive transcripts that do not have predicted canonical 3'UTR miR-488–3p MREs using biotinylated miR-488–3p as bait as compared to the control biotinylated miRNA. Error bars s.e.m. (*n* = 3 biological replicates). 2-sided *t*-test; ***p<0.001, **p<0.01, *p<0.05, ns – not significant.
DOI: https://doi.org/10.7554/eLife.45051.021
The following source data is available for figure 6:

**Source data 1.** N2A wildtype and miR-488–3 p mutant Cerox1 overexpression complex I and citrate synthase assays - *Figure 6F*.
DOI: https://doi.org/10.7554/eLife.45051.022

MRE (*Figure 6C*). As expected for a bona fide MRE, these substitutions abrogated the ability of miR-488–3p to destabilise *Cerox1* transcript in a luciferase assay (*Figure 6D*). Importantly, these substitutions also abolished the ability of *Cerox1*, when overexpressed, to elevate complex I transcript levels (*Figure 6E*), and to enhance complex I enzymatic activity (*Figure 6F*). The latter observation is important because not all bona fide miRNA-transcript interactions are physiologically active (*Bassett et al., 2014*).

Finally, direct physical interaction between *Cerox1* and miR-488–3p was confirmed by pulling-down transcripts with biotinylated miR-488–3p (*Figure 6G*). This experiment also identified 10 of 12 complex I transcripts tested as direct targets of miR-488–3p binding. These included transcripts not predicted as containing a miR-488–3p MRE, as expected from the high false negative rate of MRE prediction algorithms (*Mazière and Enright, 2007*; *Yue et al., 2009*; *Tabas-Madrid et al., 2014*). Also as expected, the two negative control transcripts, which are not responsive to *Cerox1* transcript levels and have no predicted MREs for miR-488–3p, failed to bind miR-488–3p.

Considered together, these findings indicate that: (i) *Cerox1* can post-transcriptionally regulate OXPHOS enzymatic activity as a miRNA decoy, and (ii) of 12 miR-488–3p:*Nduf* transcript interactions that were investigated, all 12 are substantiated either by responsiveness to miR-488–3p through miRNA overexpression or inhibition (*Figure 6A,B*), or by direct interaction with a biotinylated miR-488–3p mimic (*Figure 6G*). Consequently, our data demonstrates that miR-488–3p directly regulates the transcript levels of *Cerox1* and at least 12 nuclear encoded mitochondrial complex I subunit genes (31% of all) and indirectly modulates complex I activity (*Figure 6F*) in N2A cells.

## *Cerox1* is an evolutionarily conserved regulator of mitochondrial complex I activity

Fewer than 20% of lncRNAs are conserved across mammalian evolution (*Necsulea et al., 2014*) and even for these functional conservation has rarely been investigated. In our final set of experiments we demonstrated that *CEROX1*, the orthologous human transcript, is functionally equivalent to mouse *Cerox1* in regulating mitochondrial complex I activity. Similar to mouse *Cerox1*, human *CEROX1* is highly expressed in brain tissue, is otherwise ubiquitously expressed (*Figure 1—figure supplement 1B*), and is enriched in the cytoplasm of human embryonic kidney (HEK293T) cells (*Figure 7A*). *CEROX1* is expressed in human tissues at unusually high levels: it occurs among the top 0.3% of all expressed lncRNAs (*Figure 7B*) and its average expression is higher than 87.5% of all protein coding genes (*GTEx Consortium, 2017*). Its expression is highest within brain tissues, particularly within the basal ganglia and cortex (*Figure 7C*).

Importantly, mitochondrial complexes' I and III activities increased significantly following *CEROX1* overexpression in HEK293T cells (*Figure 7D*). *CEROX1* overexpression had a greater effect on complex I activity than the mouse orthologous sequence and also increased the activity of

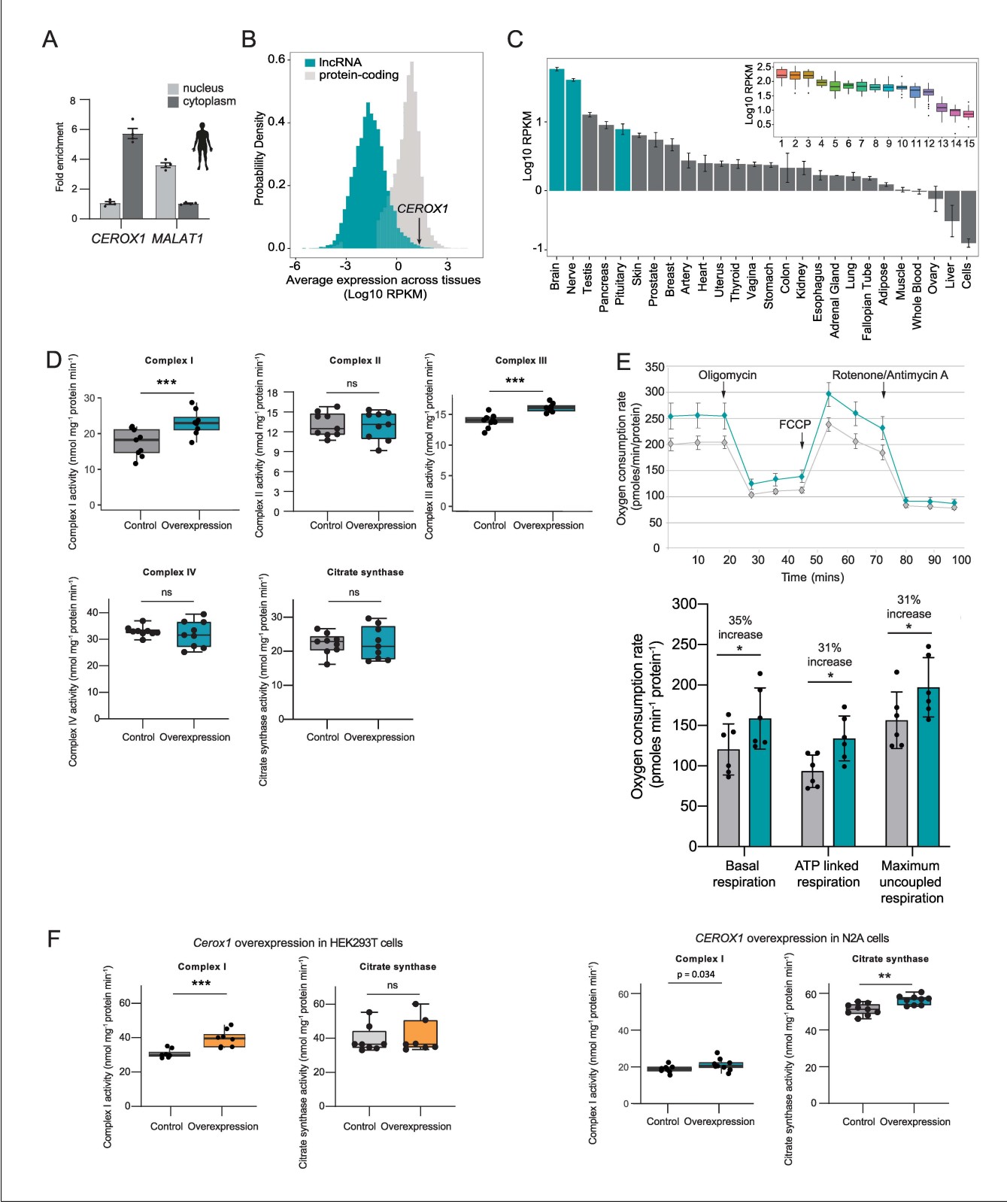

**Figure 7.** Human *CEROX1* modulates complex I activity in mouse cells. (**A**) *CEROX1* is enriched in the cytoplasm. Error bars s.e.m. *n* = 4 biological replicates. (**B**) Relative levels of lncRNA (blue) and protein-coding gene (grey) expression across individuals and tissues in human. The black arrow indicates the expression level of *CEROX1* in the set of 5161 lncRNAs. RPKM: reads per kilobase per million reads. (**C**) Average expression levels of *CEROX1* in human tissues. Blue bars highlight neurological tissues used to build the inset graph. The inset graphic represents the comparison of gene

*Figure 7 continued on next page*

Figure 7 continued

expression variation among individuals for neurological tissues: 1–Putamen, 2-Caudate nucleus, 3-Nucleus accumbens, 4-Cortex, 5-Substantia nigra, 6–Amygdala, 7–Hippocampus, 8-Spinal cord, 9-Anterior cingulate cortex, 10-Frontal cortex, 11–Hypothalamus, 12-Tibial nerve, 13–Cerebellum, 14–Pituitary gland, 15-Cerebellar hemisphere. (D) OXPHOS enzyme activities in human HEK293T cells after 72 hr of *CEROX1* overexpression. Overexpression of *CEROX1* results in significant increases in the activities of complexes I (31% increase) and III (18% increase), with no significant change in other enzyme activities. *n* = 8 biological replicates. 2-sided *t*-test: p<0.01, ns = not significant. (E) Oxygen consumption, as measured on a Seahorse XF$^e$24 Analyzer, by HEK293T cells overexpressing *CEROX1*. Top: normalised real time oxygen consumption rate in basal conditions and after sequential injections of oligomycin, FCCP and rotenone/antimycin A. Bottom: changes in basal, ATP-linked and maximum uncoupled respiration respectively. Error bars s.e.m. *n* = 6 biological replicates. (F) Reciprocal overexpression of mouse *Cerox1* in human HEK293T cells or human *CEROX1* in mouse N2A cells results in elevated complex I activity. *n* = 8 biological replicates. 2-sided *t*-test: ***p<0.001, **p<0.01, ns = not significant.
DOI: https://doi.org/10.7554/eLife.45051.023

The following source data is available for figure 7:

**Source data 1.** HEK293T CEROX1 overexpression specific enzyme assays - *Figure 7D*.
DOI: https://doi.org/10.7554/eLife.45051.024
**Source data 2.** HEK293T CEROX1 overexpression seahorse bioanalyzer - *Figure 7E*.
DOI: https://doi.org/10.7554/eLife.45051.025
**Source data 3.** Reciprocal overexpression, complex I and citrate synthase assay - *Figure 7F*.
DOI: https://doi.org/10.7554/eLife.45051.026

complex III, rather than complex IV activity, in these cells. In addition to these observed increases in enzyme activity, basal respiration increased by 35%, ATP-linked respiration increase by 31% and maximum uncoupled respiration increased by 31% (p=0.02, p=0.04, p=0.01 respectively, 2-tailed Student's *t*-test; *Figure 7E*). The latter distinction could reflect the differences in miRNA pools between mouse and human cell lines and/or the presence of different MREs in the lncRNA and human OXPHOS transcripts.

Strikingly, either reciprocal expression of mouse *Cerox1* in human HEK293T cells or human *CEROX1* in mouse N2A cells, recapitulates the previously observed increase in complex I activity (*Figure 7F*). This effect of mouse *Cerox1* overexpression in mouse N2A cells is greater than for human *CEROX1* overexpression in these cells. The role of both *Cerox1* and *CEROX1* in modulating the activity of mitochondrial complex I has thus been conserved over 90 million years since the last common ancestor of mouse and human.

## Discussion

*Cerox1* is the first evolutionarily conserved lncRNA to our knowledge that has been demonstrated experimentally to regulate mitochondrial energy metabolism. Its principal location in N2A cells is in the cytoplasm (*Figure 1F*) where it post-transcriptionally regulates the levels of mitochondrial OXPHOS subunit transcripts and proteins by decoying for miRNAs (*Figure 5—figure supplement 1*), most particularly miR-488–3p. This microRNA shares with *Cerox1* an early eutherian origin and elevated expression in brain samples (*Landgraf et al., 2007*), and it previously was shown to alter mitochondrial dynamics in cancer cells (*Yang et al., 2017*). Changes in *Cerox1* abundance in vitro alter mitochondrial OXPHOS subunit transcript levels and, more importantly, elicit larger changes in their protein subunits levels, leading to unexpectedly large changes in mitochondrial complex I catalytic activity. The observed changes in catalytic activity are in line with the degree of change seen in diseases exhibiting mitochondrial dysfunction (*Schapira et al., 1990*; *Ritov et al., 2005*; *Andreazza et al., 2010*). Overexpression of *Cerox1* in N2A cells increases oxidative metabolism, halves cellular oxidative stress and enhances protection against the complex I inhibitor rotenone. The effect of *Cerox1* on complex I subunit transcript levels can be explained by their sharing MREs with *Cerox1*, and subsequent competition for miRNA binding, most notably for miR-488–3p, which buffers the OXPHOS transcripts against miRNA-mediated repression.

Multiple RNA transcripts have been experimentally shown to compete with mRNAs for binding to miRNAs, thereby freeing the protein coding mRNA from miRNA-mediated repression (*Cesana et al., 2011*; *Karreth et al., 2011*; *Sumazin et al., 2011*; *Tay et al., 2011*; *Karreth et al., 2015*; *Tan et al., 2014*). It has been experimentally demonstrated that this miRNA:RNA regulatory crosstalk can initiate rapid co-ordinate modulation of transcripts whose proteins participate within

the same complex or process (*Tan et al., 2014*). Physiological relevance of this crosstalk mechanism remains incompletely understood. Furthermore, mathematical modelling (*Ala et al., 2013*; *Figliuzzi et al., 2013*; *Martirosyan et al., 2016*) and experimental investigation (*Bosson et al., 2014*; *Denzler et al., 2014*) of the dynamics and mechanism of endogenous transcript competition for miRNA binding have resulted in contrasting conclusions. Current mathematical models do not take full account of miRNA properties, such as the repressive effect not being predictable from its cellular abundance (*Mullokandov et al., 2012*), intracellular localisation such as at the rough ER (*Stalder et al., 2013*), loading on the RNA-induced silencing complex (RISC) (*Flores et al., 2014*), or AGO2's phosphorylation status within the RISC (*Golden et al., 2017*). The conclusions of experiments have also assumed that all miRNA target transcripts that contain the same number and affinity of miRNA binding sites are equivalent, that steady-state measurements are relevant to repression dynamics, and that observations for one miRNA in one experimental system are equally applicable to all others (*Smillie et al., 2018*).

Considered together, our lines of experimental evidence indicate that miRNA-mediated target competition by *Cerox1* substantially perturbs a post-transcriptional gene regulatory network that includes at least 12 complex I subunit transcripts. This is consistent with the expression level of miR-488–3p (*Kozomara and Griffiths-Jones, 2014*) and the high in vivo expression of both *Cerox1* and OXPHOS transcripts (*Schwanhäusser et al., 2011*; *Vogel et al., 2010*). Human *CEROX1* levels, for example, exceed those of all complex I subunit transcripts (those in *Figure 6a*) in both newly-formed and myelinating oligodendrocytes (*Forrest et al., 2014*). *Cerox1* could maintain OXPHOS homeostasis in cells with sustained high metabolic activity and high energy requirements. Such cells occur in the central nervous system, in which *Cerox1* levels are high (*Sokoloff, 1977*), and in haematopoiesis where depletion of *Cerox1* results in B cell depletion or myeloid enrichment (*Delás et al., 2019*).

Our experiments demonstrate that post-transcriptional regulation of a subset of complex I subunits by *Cerox1* leads to elevated oxygen consumption. How consumption increases when there is a higher abundance of only a subset of OXPHOS transcripts remains unclear. However, this phenomenon has been observed previously in mouse dopaminergic neurons (*Alvarez-Fischer et al., 2011*) and primary mouse embryonic fibroblasts and pinnal tissues (*Shyh-Chang et al., 2013*). Our observation of increased enzymatic activity may relate to the formation, by the complexes of the respiratory chain, of higher order supercomplexes (*Schägger and Pfeiffer, 2000*; *Genova and Lenaz, 2014*). Alternatively, the observed increases in OXPHOS activity may reflect some subunits of the complex I holo-enzyme (including NDUFS3 and NDUFA2) being present as a monomer pool and therefore being available for direct exchange without integration into assembly intermediates (*Dieteren et al., 2012*). This monomer pool facilitates the rapid swapping out of oxidatively damaged complex I subunits (*Dieteren et al., 2012*). It is thus possible that *Cerox1*-mediated expansion of the monomer pool thereby improves complex I catalysis efficiency (*Figure 8*). However, we note that overexpression of *Cerox1* results in the differential expression of 286 genes, most (83%) of which are upregulated. These genes' transcripts will be both the targets of miR-488–3p decoying by *Cerox1* (*Figure 6*) and those whose upregulation is secondary to *Cerox1* (and miR-488–3p) mediated effects, for example relating to the observed changes in cellular metabolism and proliferation (*Figure 2—figure supplement 1I,J,K*).

More efficient ETC enzymatic activity might be relevant to mitochondrial dysfunction, a feature of many disorders that often manifests as decreases in the catalytic activities of particular mitochondrial complexes. A decrease in catalytic activity can result in elevated ROS production, leading to oxidative damage of lipids, DNA, and proteins, with OXPHOS complexes themselves being particularly susceptible to such damage (*Musatov and Robinson, 2012*). Parkinson's and Alzheimer's diseases both feature pathophysiology associated with oxidative damage resulting from increased ROS production and a cellular energy deficit associated with decreased complex I and IV activities (a reduction of 30% and 40%, respectively) (*Schapira et al., 1990*; *Keeney et al., 2006*; *Canevari et al., 1999*). A deficiency in complexes II, III and to a lesser extent complex IV, has also been described in Huntington disease (*Mochel and Haller, 2011*). Currently no effective treatments exist that help to restore mitochondrial function despite demonstration that a 20% increase in complex I activity protects mouse midbrain dopaminergic neurons against MPP+, a complex I inhibitor and a chemical model of Parkinson's disease (*Alvarez-Fischer et al., 2011*). We note that the highest expression of *CEROX1* occurs primarily in the basal ganglia (*Figure 7C* inset) regions of which are specifically vulnerable to the progressive neurological disorders Parkinson's (substantia nigra pars compacta) and

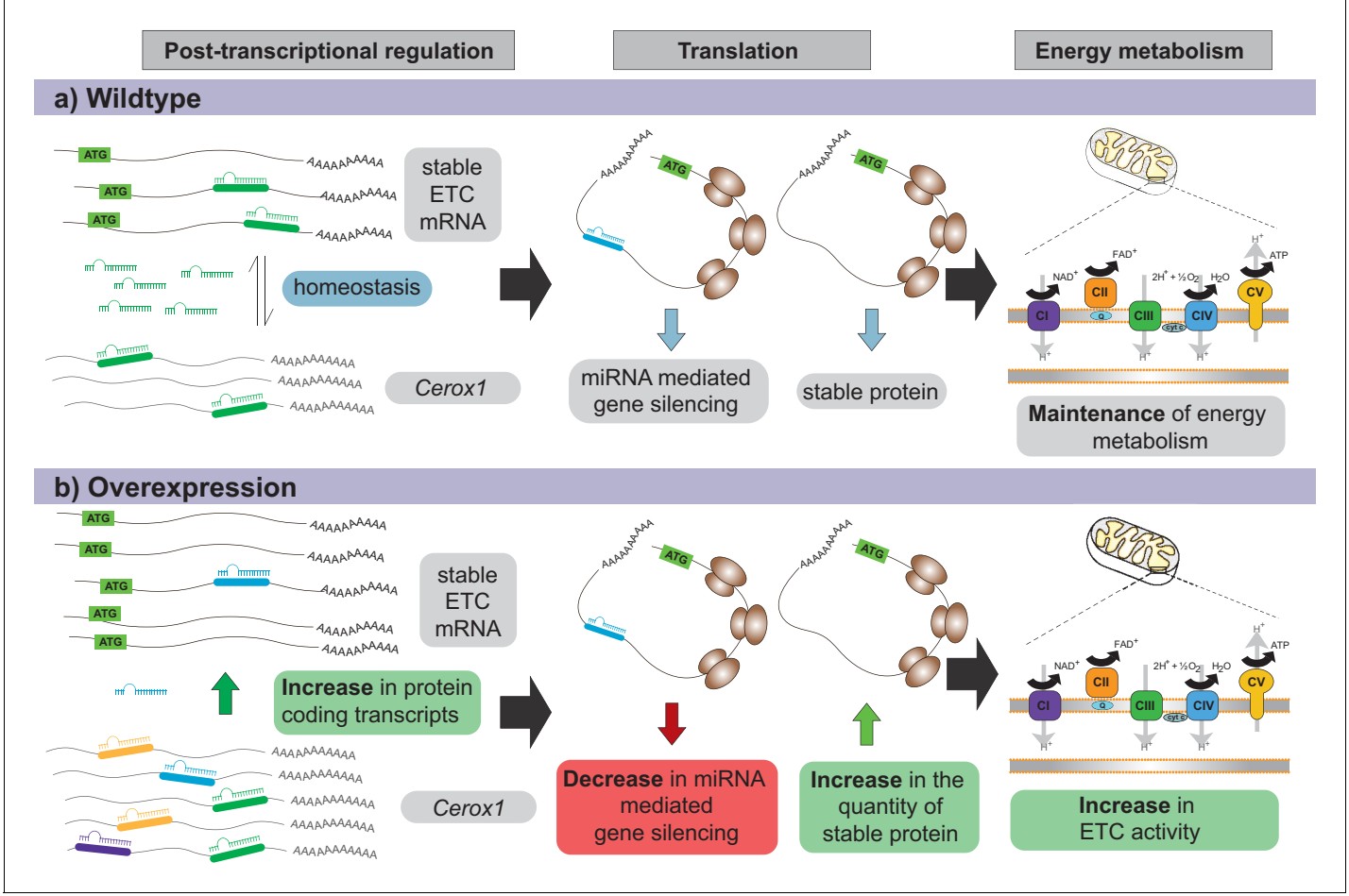

**Figure 8.** Proposed model for *Cerox1* as a post-transcriptional regulator of mitochondrial protein production and energy metabolism. In this model, *Cerox1* (**A**) post-transcriptionally maintains energy metabolism homeostasis through buffering the stable ETC transcripts against miRNA-mediated gene silencing. Overexpression of *Cerox1* (**B**) leads to a depletion of the pool of miRNAs that bind ETC transcripts, and therefore a decrease in miRNA mediated gene silencing of the ETC protein-coding transcripts. This has two subsequent effects: (1) a further accumulation of ETC protein coding transcripts, and (2) an increase in the overall translation of ETC subunit proteins owing to decreased miRNA binding to ETC transcripts. More rapid replenishment by undamaged subunits in mitochondrial complex I, leads to increased efficiency of complex I activity and hence an increase in overall oxygen consumption of the ETC.

DOI: https://doi.org/10.7554/eLife.45051.027

Huntington's diseases (striatum: caudate and putamen). The specific energy demands of these neurons may make them particularly susceptible to damage due to an energy deficit. For instance, the dopaminergic neurons of the substantia nigra, which are especially sensitive to degeneration in Parkinson's disease, have unusually large axonal arbours that require tight regulation of cellular energy to maintain (*Bolam and Pissadaki, 2012*). In addition, the medium spiny neurons of the striatum, which preferentially degenerate in Huntington disease, exhibit a high degree of axonal collateralization – a morphological trait which implies high cellular energy consumption for its maintenance (*Parent and Parent, 2006*) – therefore causing these cells to be vulnerable to decreased cellular ATP production. *CEROX1*'s ability to increase mitochondrial complex I activity might be recapitulated pharmacologically to restore mitochondrial function, as an exemplar of therapeutic upregulation of gene expression (*Wahlestedt, 2013*).

## Materials and methods

**Key resources table**

| Reagent type (species) or resource | Designation | Source or reference | Identifiers | Additional information |
|---|---|---|---|---|
| Gene (*Mus musculus*) | *Cerox1* | NA | AK079380; 2810468N07Rik; ENSMUST00000163493 | |
| Gene (*Homo sapiens*) | *CEROX1* | NA | BC098409; RP11-161M6.2; ENST00000562570 | |
| Cell line (*Mus musculus*) | N2A (mouse neuroblastoma cells) | European Collection of Authenicated Cell Cultures (ECACC) | RRID: CVCL_0470; ECACC: 89121404 | |
| Cell line (*Homo sapiens*) | HEK293T (human embryonic kidney cells) | European Collection of Authenicated Cell Cultures (ECACC) | RRID: CVCL_0063; ECACC 12022001 | |
| Transfected construct (*Mus musculus* and *Homo sapiens*) | pCAG-GFP | https://www.addgene.org/89684/ | RRID:Addgene_89684 | This backbone was modified for overexpression of *Cerox1*, *Cerox1* MRE mutants and *CEROX1* |
| Transfected construct (*Mus musculus*) | BLOCK-it U6 shRNA expression construct | Invitrogen | K494500 | |
| Transfected construct (*Mus musculus*) | BLOCK-iT Pol II miR RNAi expression vector | Invitrogen | K493600 | |
| Antibody | anti-NDUFS1, rabbit monoclonal | Abcam | RRID: AB_2687932; ab169540 | Overnight, four degrees (1:30,000) |
| | anti-NDUFS3, mouse monoclonal | Abcam | RRID:AB_10861972; ab110246 | Overnight, four degrees, 0.15 mg/ml |
| | anti-alpha tubulin, mouse monoclonal | Abcam | RRID:AB_2241126; ab7291 | Overnight, four degrees (1:30,000) |
| | goat anti-rabbit HRP, goat polyclonal | Invitrogen | RRID:AB_2536530; G-21234 | Room temperature, 1 hr (1:30,000) |
| | goat anti-mouse HRP, goat polyclonal | Dako | RRID:AB_2617137; P0447 | Room temperature, 1 hr (1:3,000) |
| Recombinant DNA reagent | Fugene6 | Promega | E2691 | |
| Commercial assay or kit | Amplex Red Hydrogen Peroxide/Peroxidase Assay Kit | Thermofisher | A22188 | |
| | OxyBlot protein oxidation detetion kit | Merck Millipor | S7150 | |
| | QuantiGene ViewRNA miRNA ISH cell assay kit | Affymetrix | QVCM0001 | |
| Chemical compound, drug | Rotenone | Sigma Aldrich | R8875-5G | |
| | Antimycin A | Sigma Aldrich | A8674-25MG | |
| | Sodium Azide | Sigma Aldrich | S8032-25G | |
| | Oligomycin | Sigma Aldrich | 75351–5 MG | |
| | NADH (reduced) | Sigma Aldrich | N8129-1G | |
| | coenzyme Q | Sigma Aldrich | C7956-2mg | |
| | Fatty acid free albumin | Sigma Aldrich | A8806-1G | |

*Continued on next page*

*Continued*

| Reagent type (species) or resource | Designation | Source or reference | Identifiers | Additional information |
|---|---|---|---|---|
| | Sodium succinate | Sigma Aldrich | S5047-100G | |
| | Dichloropheno lindophenol (DCPIP) | Sigma Aldrich | 33125–5 G-R | |
| | Decylubiquinone | Sigma Aldrich | D7911-10mg | |
| | cytochrome c | Sigma Aldrich | C7752 | |
| | Carbonyl cyanide 4(trifluoromethoxy) phenylhydrazone | Sigma Aldrich | C2920-10mg | |
| | miRNA inhibitors | Ambion | 4464084 | |
| | miRCURY LNA biotinylated miRNAs | Exiqon | 339178 | |
| Software, algorithm | TargetScan v7.0 | *Agarwal et al., 2015* | DOI: 10.7554/eLife.05005 | |

## Gene expression profiling

The lncRNA transcripts were assessed for coding potential using the coding potential calculator (*Kong et al., 2007*), PhyloCSF (*Lin et al., 2011*) and by mining proteomics and small open reading frame resources for evidence of translation (*Wilhelm et al., 2014*; *Kim et al., 2014a*; *Bazzini et al., 2014*). A lack of protein-coding potential for human *CEROX1* (also known as RP11-161M6.2, LMF1-3) is supported by a variety of computational and proteomic data summarised in LNCipedia (*Volders et al., 2015*). Expression data from somite-staged mouse embryos were acquired from ArrayExpress (E-ERAD-401 - Strand-specific RNA-seq of somite-staged second generation genotypically wild-type embryos of mixed G0 lineage from the Mouse Genetics Project/DMDD). Genome wide associations were performed on the UK Biobank data as described in *Canela-Xandri et al. (2018)* using data from up to 452,264 individuals.

Total RNA from twenty normal human tissues (adipose, bladder, brain, cervix, colon, oesophagus, heart, kidney, liver, lung, ovary, placenta, prostate, skeletal muscle, small intestine, spleen, testes, thymus, thyroid and trachea) were obtained from FirstChoice Human Total RNA Survey Panel (Invitrogen). Total RNA from twelve mouse tissues (bladder, brain, colon, heart, kidney, liver, pancreas, skeletal muscle, small intestine, stomach and testis) were obtained from Mouse Tissue Total RNA Panel (Amsbio). RNA from cell lines was extracted using the RNeasy mini kit (Qiagen) according to the manufacturer's instructions, using the optional on column DNase digest. cDNA synthesis for all samples was performed on 1 µg of total RNA using a QuantiTect Reverse Transcription kit (Qiagen) according to the manufacturer's instructions. RNA was extracted from samples used for the detection of miRNAs using the miRNeasy mini kit (Qiagen) according to the manufacturer's instructions (with on column DNase digest). All RNA samples were quantified using the 260/280 nm absorbance ratio, and RNA quality assessed using a Tapestation (Agilent). RNA samples with an RNA integrity number (RIN) >8.5 were reverse transcribed. 1 µg of total RNA from the miRNA samples were reverse transcribed using the NCode VILO miRNA cDNA synthesis kit. Expression levels were determined by real-time quantitative PCR, using SYBR Green Master Mix (Applied Biosystems) and standard cycling parameters (95°C 10 min; 40 cycles 95°C 15 s, 60°C 1 min) followed by a melt curve using a StepOne thermal cycler (Applied Biosystems). All amplification reactions were performed in triplicate using gene specific primers. Multiple reference genes were assessed for lack of variability using geNorm (*Vandesompele et al., 2002*). Human expression data were normalised to *TUBA1A* and *POLR2A*, whilst mouse expression data were normalised to *Tbp* and *Polr2a*. Oligonucleotide sequences are provided in *Supplementary file 4*.

## Tissue culture and flow cytometry

Mouse Neuro-2a neuroblastoma cells (N2A; RRID: CVCL_0470; ECACC 89121404) and human embryonic kidney (HEK293T; RRID: CVCL_0063; ECACC 12022001) were sourced from the European authenticated cell culture collection. HEK293T cells were confirmed by STR profiling and cell lines were tested monthly for mycoplasma contamination. Cells were grown at 37°C in a humidified

incubator supplemented with 5% $CO_2$. Both cell lines were grown in Dulbecco's modified Eagle medium containing penicillin/streptomycin (100 U/ml, 100 ug/ml respectively) and 10% fetal calf serum. Cells were seeded at the following densities: six well dish, $0.3 \times 10^6$; 48 well dish, $0.2 \times 10^4$; T75 flask $2.1 \times 10^6$. We had three reasons for the choice of HEK293T cells for this experiment. First, they are of neural crest ectodermal origin (*Lin et al., 2014*) and therefore have a number of neural cell line characteristics in that they express neuronal markers (*Shaw et al., 2002*). Second, they are in use as a cell culture model for neurodegenerative diseases such as Parkinson's disease (*Falkenburger et al., 2016*; *Schlachetzki et al., 2013*). Third, HEK293T cells are a widely used cell line to interrogate human mitochondrial biochemistry, and exhibit complex I dependent respiration (*Kim et al., 2014b*). Mouse embryonic stem cells and dicer knock-out embryonic stem cells were maintained as described previously (*Nesterova et al., 2008*). Cells were counted using standard haemocytometry. For flow cytometry the cells were harvested by trypsinization, washed twice with PBS and fixed in 70% ethanol (filtered, −20°C). The cell suspension was incubated at 4°C for 10 min and the cells pelleted, treated with 40 µg/ml RNase A and propidium iodide (40 µg/ml) for 30 min at room temperature. Cells were analysed using a FACSCalibur (BD-Biosciences) flow cytometer.

## Fluorescent in situ hybridization of miRNA and lncRNA, cellular fractionation and RNA turnover

Branched chain DNA probes to *Cerox1, Malat1* and mmu-miR-488–3p were sourced from Affymetrix. The protocol was preformed according to the manufacturer's instructions using the QuantiGene ViewRNA miRNA ISH cell assay kit (QVCM0001) for adherent cells. The following parameters were optimised: cells were fixed in 4% formaldehyde for 45 min and a 1:2000 dilution of the protease was optimal for the N2A cells. Cells were imaged using an Andor Dragonfly confocal inverted microscope, and images acquired using an Andor Zyla 4.2 plus camera.

Cells were fractionated into nuclear and cytoplasmic fractions in order to determine the predominant cellular localization of lncRNA transcripts. Briefly, approximately $2.8 \times 10^6$ cells were collected by trypsinization, washed three times in PBS and pelleted at 1000 g for 5 min at 4°C. The cell pellet was resuspended in 5 volumes of lysis buffer (10 mM Tris-HCl, pH 7.5, 3 mM $MgCl_2$, 10 mM NaCl, 5 mM EGTA, 0.05% NP40, and protease inhibitors [Roche, complete mini]) and incubated on ice for 15 min. Lysed cells were then centrifuged at 2000 g for 10 min at 4°C, and the supernatant collected as the cytoplasmic fraction. Nuclei were washed three times in nuclei wash buffer (10 mM HEPES, pH 6.8, 300 mM sucrose, 3 mM $MgCl_2$ 25 mM NaCl, 1 mM EGTA), and pelleted by centrifugation at 400 g, 1 min at 4°C. Nuclei were extracted by resuspension of the nuclei pellet in 200 µl of nuclei wash buffer containing 0.5% Triton X-100 and 700 units/ml of DNase I and incubated on ice for 30 mins. Nucleoplasm fractions were collected by centrifugation at 17 000 g for 20 min at 4°C. RNA was extracted as described above, and RNA samples with RIN values > 9.0 used to determine transcript localisation.

To determine the stability of the lncRNA transcripts, cells were cultured to ~50% confluency and then transcription was inhibited by the addition of 10 µg/ml actinomycin D (Sigma) in DMSO. Control cells were treated with equivalent volumes of DMSO. Transcriptional inhibition of the N2A cells was conducted for 16 hr with samples harvested at 0 hr, 30 mins, 1 hr, 2 hr, 4 hr, 8 hr and 16 hr. RNA samples for fractionation and turnover experiments were collected in Trizol (Invitrogen) and RNA purified and DNAse treated using the RNeasy mini kit (Qiagen). Reverse transcription for cellular localisation and turnover experiments was performed as described earlier.

## Constructs and biotinylated miRNAs

The 5' and 3' ends of *Cerox1* and *CEROX1* were identified by 5' and 3' RACE using the GeneRacer Kit (Invitrogen) according to manufacturer's instructions. As an overexpression/transfection control the pCAG-EGFP backbone was used (RRID:Addgene_89684). The EGFP was removed from this backbone, and all full length lncRNAs were cloned into the pCAG backbone. For cloning into the pCAGs vector, PCR primers modified to contain the cloning sites BglII and XhoI sites were used to amplify the full length mouse *Cerox1*, whilst human *CEROX1* and the mouse 5x MRE mutant were synthesized by Biomatik (Cambridge, Ontario), and also contained BglII and XhoI sites at the 5' and 3' ends respectively. All other MRE mutants were produced using overlapping PCR site directed mutagenesis to mutate 3 bases of the miRNA seed region. All purified products were ligated into

the prepared backbone and then transformed by heat shock into chemically competent DH5α, and plated on selective media. All constructs were confirmed by sequencing. Short hairpin RNAs specific to the transcripts were designed using a combination of the RNAi design tool (Invitrogen) and the siRNA selection program from the Whitehead Institute (*Yuan et al., 2004*). Twelve pairs of shRNA oligos to the target genes and β-galactosidase control oligos were annealed to create double-stranded oligos and cloned into the BLOCK-iT U6 vector (Invitrogen), according to the manufacturer's instructions. miRNA expression constructs were generated and cloned into the BLOCK-iT Pol II miR RNAi expression vector (Invitrogen) according to the manufacturer's instructions. miRNA inhibitors were sourced from Ambion and used according to manufacturer's instructions.

Transfection efficiency was initially assessed by FACS, and the optimised transfection protocol used for all further assays (6:1 transfection reagent to DNA ratio). One day prior to transfection cells were either seeded in six well dishes ($0.3 \times 10^6$ cells/well), or in T75 flasks ($2.1 \times 10^6$ cells/flask). Twenty-four hours later cells in six well dishes were transfected with 1 μg of shRNA, miRNA or overexpression construct and their respective control constructs using FuGENE 6 (Promega) according to the manufacturer's guidelines. Cells in T75 flasks were transfected with 8 μg of experimental or control constructs. Transfected cells were grown for 48-72 hrs under standard conditions, and then harvested for either gene expression studies or biochemical characterisation. Efficacy of the overexpression and silencing constructs was determined by real-time quantitative PCR.

Transcripts for the luciferase destabilisation assays were cloned into the pmirGLO miRNA target expression vector (Promega) and assayed using the dual-luciferase reporter assay system (Promega). miRCURY LNA biotinylated miRNAs (mmu-miR-488–3p and mmu-negative control 4) were purchased from Exiqon, and direct mRNA-miRNA interactions were detected using a modified version of *Orom and Lund (2007)* and enrichment of targets was detected by qPCR.

## Computational techniques

MREs were predicted using TargetScan v7.0 (*Agarwal et al., 2015*; RRID: SCR_010845) in either the 3'UTR (longest annotated UTR, ENSEMBL build 70; RRID: SCR_002344) or the full length transcript of protein coding genes, and across the entire transcript for lncRNAs. The average expression across 46 human tissues and individuals according to the Pilot one data from the GTEx Consortium (*Lonsdale et al., 2013*; RRID:SCR_013042; 98) was computed for both protein-coding genes and intergenic lncRNAs from the Ensembl release 75 annotation (*Flicek et al., 2014*). We used the normalised number of CAGE tags across 399 mouse cells and tissues from the FANTOM5 Consortium (http://fantom.gsc.riken.jp; *Kawaji et al., 2014*) as an approximation of expression levels for protein-coding genes and intergenic lncRNAs from the Ensembl release 75 annotation. If multiple promoters were associated with a gene, we selected the promoter with the highest average tag number. Conserved sequence blocks in the lncRNA sequences were identified using LALIGN (*Goujon et al., 2010*).

## Microarray analysis

Microarray analysis was performed on 16 samples (four overexpression/four overexpression controls; four knock-down/four knock-down controls), and hybridizations were performed by the OXION array facility (University of Oxford). Data were analysed using the web-based Bioconductor interface, CARMAweb (*Rainer et al., 2006*). Differentially expressed genes (Bonferroni corrected p-value<0.05) were identified between mouse lncRNA overexpression and control cells using Limma from the Bioconductor package between the experimental samples and the respective controls. Microarray data are accessible through ArrayExpress, accession E-MATB-6792.

## Extraction of metabolites and liquid chromatography – mass spectrometry (LC-MS)

Metabolites were extracted from 6-well plates by washing individual wells with ice-cold PBS and addition of cold extraction buffer (50% methanol, 30% acetonitrile, 20% water solution at −20°C or lower). Extracts were clarified and stored at −80°C until required. LC-MS was carried out using a 100 mm x 4.6 mm ZIC-pHILIC column (Merck-Millipore) using a Thermo Ultimate 3000 HPLC inline with a Q Exactive mass spectrometer. A 32 min gradient was developed over the column from 10% buffer A (20 mM ammonium carbonate), 90% buffer B (acetonitrile) to 95% buffer A, 5% buffer B. 10 μl of

metabolite extract was applied to the column equilibrated in 5% buffer A, 95% buffer B. Q Exactive data were acquired with polarity switching and standard ESI source and spectrometer settings were applied (typical scan range 75–1050). Metabolites were identified based upon m/z values and retention time matching to standards. Quantitation of metabolites was carried out using AssayR (*Wills et al., 2017*). Data were normalised using levels of 9 essential amino acids (histidine, isoleucine, leucine, lysine, methionine, phenylalanine, threonine, tryptophan, valine), and errors propagated, in order to account for cell count differences.

## Western blots

Total protein was quantified using a BCA protein assay kit (Pierce). 10 µg of protein was loaded per well, and samples were separated on 12% SDS-PAGE gels in Tris-glycine running buffer (25 mM Tris, 192 mM glycine, 0.1% SDS). Proteins were then electroblotted onto PVDF membrane (40V, 3 hr) in transfer buffer (25 mM Tris-HCl, 192 mM glycine, 20% methanol), the membrane blocked in TBS-T (50 mM Tris-HCl, 150 mM NaCl, 0.1% Tween 20) with 5% non–fat milk powder for 1 hr. The membrane was incubated with primary antibodies overnight at 4°C with the following dilutions: anti-NDUFS1 (RRID: AB_2687932; rabbit monoclonal, ab169540, 1:30,000), anti-NDUFS3 (RRID:AB_10861972; mouse monoclonal, 0.15 mg/ml, ab110246), or anti-alpha tubulin loading control (RRID: AB_2241126; mouse monoclonal, ab7291, 1:30,000). Following incubation with the primary antibodies, blots were washed 3 × 5 min, and 2 × 15 mins in TBS-T and incubated with the appropriate secondary antibody for 1 hr at room temperature: goat anti-rabbit HRP (RRID:AB_2536530; Invitrogen G-21234) 1:30,000; goat anti-mouse HRP (RRID:AB_2617137; Dako P0447) 1:3000. After secondary antibody incubations, blots were washed and proteins of interested detected using ECL prime chemiluminescent detection reagent (GE Healthcare) and the blots imaged using an ImageQuant LAS 4000 (GE Healthcare). Signals were normalised to the loading control using ImageJ (*Schneider et al., 2012*).

## Oxidative phosphorylation enzyme assays and oxygen consumption

Cell lysates were prepared 48 hr post-transfection, by harvesting cells by trypsinisation, washing three times in ice cold phosphate buffered saline followed by centrifugation to pellet the cells (two mins, 1000 g). Cell pellets were resuspended to homogeneity in KME buffer (100 mM KCl, 50 mM MOPS, 0.5 mM EGTA, pH 7.4) and protein concentrations were determined using a BCA protein assay detection kit (Pierce). Cell lysates were flash frozen in liquid nitrogen, and freeze-thawed three times prior to assay. 300–500 µg of cell lysate was added per assay, and assays were normalised to the total amount of protein added.

All assays were performed using a Shimadzu UV-1800 spectrophotometer, absorbance readings were taken every second and all samples were measured in duplicate. Activity of complex I (CI, NADH:ubiquinone oxidoreductase) was determined by measuring the oxidation of NADH to $NAD^+$ at 340 nm at 30°C in an assay mixture containing 25 mM potassium phosphate buffer (pH 7.2), 5 mM $MgCl_2$ 2.5 mg/ml fatty acid free albumin, 0.13 mM NADH, 65 µM coenzyme Q and 2 µg/ml antimycin A. The decrease in absorbance was measured for 3 mins, after which rotenone was added to a final concentration of 10 µM and the absorbance measured for a further 2 mins. The specific complex I rate was calculated as the rotenone-sensitive rate minus the rotenone-insensitive rate. Complex II (CII, succinate dehydrogenase) activity was determined by measuring the oxidation of DCPIP at 600 nm at 30°C. Lysates were added to an assay mixture containing 25 mM potassium phosphate buffer (pH 7.2) and 2 mM sodium succinate and incubated at 30°C for 10 mins, after which the following components were added, 2 µg/ml antimycin A, 2 µg/ml rotenone, 50 µM DCPIP and the decrease in absorbance was measured for 2 mins. Complex III (CIII, Ubiquinol:cytochrome *c* oxidoreductase) activity was determined by measuring the oxidation of decylubiquinol, with cytochrome *c* as the electron acceptor at 550 nm. The assay cuvettes contained 25 mM potassium phosphate buffer (pH 7.2), 3 mM sodium azide, 10 mM rotenone and 50 µM oxidized cytochrome *c*. Decylubiquinol was synthesized by acidifying decylubiquinone (10 mM) with HCl (6M) and reducing the quinine with sodium borohydride. After the addition of 35 µM decylubiquinol, the increase in absorbance was measured for 2 mins. Activity of Complex IV (CIV, cytochrome *c* oxidase) was measured by monitoring the oxidation of cytochrome *c* at 550 nm, 30°C for 3 min. A 0.83 mM solution of reduced cytochrome *c* was prepared by dissolving 100 mg of cytochrome *c* in 10 ml of potassium

phosphate buffer, and adding sodium ascorbate to a final concentration of 5 mM. The resulting solution was added into SnakeSkin dialysis tubing (7 kDa molecular weight cutoff, Thermo Scientific) and dialyzed against potassium phosphate buffer, with three changes, at 4°C for 24 hr. The redox state of the cytochrome $c$ was assessed by evaluating the absorbance spectra from 500 to 600 nm. The assay buffer contained 25 mM potassium phosphate buffer (pH 7.0) and 50 µM reduced cytochrome $c$. The decrease in absorbance at 550 nm was recorded for 3 mins. As a control the enzymatic activity of the tricarboxylic acid cycle enzyme, citrate synthase (CS) was assayed at 412 nm at 30°C in a buffer containing 100 mM Tris-HCl (pH 8.0), 100 µM DTNB (5,5-dithiobis[2-nitrobenzoic acid]), 50 µM acetyl coenzyme A, 0.1% (w/v) Triton X-100 and 250 µM oxaloacetate. The increase in absorbance was monitored for 2 mins.

The following extinction coefficients were applied: complex I (CI), $\varepsilon$ = 6220 $M^{-1}$ $cm^{-1}$, CII, $\varepsilon$ = 21,000 $M^{-1}$ $cm^{-1}$; CIII, $\varepsilon$ = 19,100 $M^{-1}$ $cm^{-1}$; CIV, $\varepsilon$ = 21,840 $M^{-1}$ $cm^{-1}$ (the difference between reduced and oxidised cytochrome $c$ at 550 nm); CS, $\varepsilon$ = 13,600 $mM^{-1}$ $cm^{-1}$.

Cellular oxygen consumption rate was determined using the Seahorse $XF^e24$ Analyzer (Agilent). Cells were plated on poly-L-lysine coated $XF^e24$ microplates at 50,000 cells per well and incubated at 37°C and 5% $CO_2$ for 24 hr. Cells were transfected with an overexpression (pCAG-Cerox1 or pCAG-CEROX1) or the Cerox1 shRNA silencing construct and assayed 24–36 hr later. Cells were washed three times with Seahorse Assay media (Agilent), supplemented with 10 mM glucose and 2 mM pyruvate. After 30 min of pre-incubation in a non-$CO_2$ 37°C incubator, cells were entered into the analyser for oxygen consumption rate measurements. After basal respiration measurements, 1 µM of oligomycin was injected to inhibit ATP synthase, then 0.2 µM of carbonyl cyanide 4-(trifluoromethoxy)phenylhydrazone (FCCP) was injected to uncouple respiration. Finally 1 µM each of rotenone and antimycin A were injected to inhibit complex I and complex III respectively. Data was normalised to total cellular protein content using the sulforhodamine B assay (*Skehan et al., 1990*). Basal respiration, ATP linked respiration and maximum uncoupled respiration were calculated from the normalised data using the Agilent Seahorse XF calculation guidelines. Briefly, basal respiration = measurement 3 (last rate measurement before oligomycin injection) – measurement 10 (minimum respiration after injection of rotenone/antimycin A); ATP-linked respiration = measurement 3 – measurement 4 (minimum rate measurement after oligomycin injection); maximum uncoupled respiration = measurement 7 (maximum rate after FCCP injection) – measurement 10.

### Oxidative stress measurements

Hydrogen peroxide production was assessed as a marker of reactive oxygen species generation using the fluorescent indicator Amplex Red (10 µM, Invitrogen) in combination with horseradish peroxidise (0.1 units $ml^{-1}$). Total amount of $H_2O_2$ produced was normalised to mg of protein added. Protein carbonylation was assessed using the OxyBlot protein oxidation detection kit (Merck Millipore), and differential carbonylation was assessed by densitometry. The cell stress assay was performed on cells seeded in 48 well plates, and assayed 12 hr later by the addition of (final concentration): rotenone (5 µM), malonate (40 µM), antimycin A (500 µM), oligomycin (500 µM), sodium azide (3 mM), NaCl (300 mM), $CaCl_2$ (5.4 mM) for 1 hr. Cells were heat shocked at 42°C and UV irradiated using a Stratlinker UV Crosslinker for 10 min (2.4 J $cm^{-2}$). Cell viability was assessed by the addition of Alamar Blue (Invitrogen) according to the manufacturer's instructions.

## Acknowledgements

This work was funded by the Wellcome Trust (106956/Z/15/Z; CPP, TMS, OBR, SRG), a European Research Council Advanced Grant (DARCGENs; CPP, ACM, TMS, KR), the Medical Research Council (CPP, WH), a Marie Curie Intra-European Career Development Award (ACM), the University of Oxford (ACM), the Royal Society (ACM), the Clarendon Fund (JYT), the Natural Sciences Engineering Research Council of Canada (JYT), and Diabetes UK (LCH, 11/0004175). RNC was funded by a Wellcome Trust Investigator award (WT100981/z/13/z) to NMM. Microarray hybridizations were performed by the OXION array facility. FISH confocal images were acquired by Matt Pearson (IGMM advanced imaging support team). We would like to thank Ava Khamseh (IGMM) for normalisation of the metabolomics data and Andrew Dodd for critical reading of the manuscript.

## Additional information

### Competing interests

Chris P Ponting: Reviewing editor, *eLife*. The other authors declare that no competing interests exist.

### Funding

| Funder | Grant reference number | Author |
|---|---|---|
| Wellcome | WT106956/Z/15/Z | Tamara M Sirey<br>Oscar Bedoya-Reina<br>Sebastian Rogatti-Granados<br>Chris P Ponting |
| European Research Council | 249869 | Tamara M Sirey<br>Kenny Roberts<br>Ana Claudia Marques<br>Chris P Ponting |
| Medical Research Council | | Wilfried Haerty<br>Chris P Ponting |
| Oxford University | The Clarendon Fund | Jennifer Y Tan |
| Natural Sciences and Engineering Research Council of Canada | | Jennifer Y Tan |
| Diabetes UK | 11/0004175 | Lisa C Heather |
| Wellcome | WT100981/z/13/z | Roderick N Carter<br>Nicholas M Morton |
| European Commission | Marie Curie Intra-European Career Development Award | Ana Claudia Marques |
| Oxford University | | Ana Claudia Marques |
| Royal Society | | Ana Claudia Marques |

The funders had no role in study design, data collection and interpretation, or the decision to submit the work for publication.

### Author contributions

Tamara M Sirey, Conceptualization, Formal analysis, Supervision, Validation, Investigation, Methodology, Writing—original draft, Writing—review and editing; Kenny Roberts, Investigation, Methodology; Wilfried Haerty, Data curation, Formal analysis; Oscar Bedoya-Reina, Formal analysis; Sebastian Rogatti-Granados, Validation, Investigation; Jennifer Y Tan, Nick Li, Investigation; Lisa C Heather, Jimi Wills, Formal analysis, Methodology; Roderick N Carter, Sarah Cooper, Nicholas M Morton, Resources, Methodology; Andrew J Finch, Resources, Formal analysis, Methodology; Ana Claudia Marques, Conceptualization, Formal analysis; Chris P Ponting, Conceptualization, Resources, Supervision, Funding acquisition, Writing—original draft, Project administration, Writing—review and editing

### Author ORCIDs

Tamara M Sirey (iD) https://orcid.org/0000-0001-5606-2858
Kenny Roberts (iD) http://orcid.org/0000-0001-6155-0821
Wilfried Haerty (iD) https://orcid.org/0000-0003-0111-191X
Sebastian Rogatti-Granados (iD) https://orcid.org/0000-0002-8438-7999
Lisa C Heather (iD) https://orcid.org/0000-0002-7246-1338
Andrew J Finch (iD) http://orcid.org/0000-0002-8065-4623
Jimi Wills (iD) https://orcid.org/0000-0003-1669-007X
Nicholas M Morton (iD) https://orcid.org/0000-0001-8218-8462

Ana Claudia Marques ![ORCID] http://orcid.org/0000-0001-5174-8092
Chris P Ponting ![ORCID] https://orcid.org/0000-0003-0202-7816

**Decision letter and Author response**
Decision letter https://doi.org/10.7554/eLife.45051.042
Author response https://doi.org/10.7554/eLife.45051.043

## Additional files

### Supplementary files

• Supplementary file 1. Association of CEROX1 single nucleotide polymorphism on anthropomorphic traits. Data was accessed through http://geneatlas.roslin.ed.ac.uk/.
DOI: https://doi.org/10.7554/eLife.45051.028

• Supplementary file 2. Differentially expressed genes after overexpression of mouse *Cerox1*.
DOI: https://doi.org/10.7554/eLife.45051.029

• Supplementary file 3. MRE predictions for Nduf subunit transcripts.
DOI: https://doi.org/10.7554/eLife.45051.030

• Supplementary file 4. shRNA oligos and PCR primers.
DOI: https://doi.org/10.7554/eLife.45051.031

• Transparent reporting form
DOI: https://doi.org/10.7554/eLife.45051.032

### Data availability

Microarray data are available through ArrayExpress, accession code E-MATB-6792.

The following dataset was generated:

| Author(s) | Year | Dataset title | Dataset URL | Database and Identifier |
|---|---|---|---|---|
| Tamara M Sirey, Chris P Ponting | 2019 | Investigation of potential gene regulatory or post-transcriptional regulatory function of a long noncoding RNA (Cerox1) | https://www.ebi.ac.uk/arrayexpress/experiments/E-MATB-6792/ | ArrayExpress, E-MATB-6792 |

The following previously published datasets were used:

| Author(s) | Year | Dataset title | Dataset URL | Database and Identifier |
|---|---|---|---|---|
| GTEx Consortium | 2013 | The Genotype-Tissue Expression (GTEx) project | https://gtexportal.org/home/datasets | GTEx Portal, phs000424.v7.p2 |
| Kawaji H, Lizio M, Itoh M, Kanamori-Katayama M, Kaiho A, Nishiyori-Sueki H, Shin JW, Kojima-Ishiyama M, Kawano M, Murata M, Ninomiya-Fukuda N, Ishikawa-Kato S, Nagao-Sato S, Noma S, Hayashizaki Y, Forrest ARR | 2014 | Comparison of CAGE and RNA-seq transcriptome profiling using clonally amplified and single-molecule next-generation sequencing | http://fantom.gsc.riken.jp/data/ | FANTOM Consortium, FANTOM5 |
| Canela-Xandri O, Rawlik K, Tenesa A | 2018 | An atlas of genetic associations in the UK Biobank. | http://geneatlas.roslin.ed.ac.uk/downloads/ | GeneATLAS, downloads |

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
