## [Decision Letter]

Thank you for submitting your work entitled "The long non-coding RNA *Cerox1* is a post transcriptional regulator of mitochondrial complex I catalytic activity" for consideration by *eLife*. Your article has been reviewed by three peer reviewers, one of who is a member of our Board of Reviewing Editors, and the evaluation has been overseen by a Senior Editor.

Our decision has been reached after consultation between the reviewers. While the reviewers found the subject matter of your manuscript is of interest and timely, they also found several issues in the manuscript that lowered their enthusiasm for the conclusions reached in the manuscript. The reviewers find the manuscript problematic in several places focused on experimental design, validation and small effect phenotypic changes supporting conclusions from targeted genetic perturbation experiments. A summary of the reviews are enclosed at the end of this letter for your review. Thus, unfortunately we are unable to recommend the acceptance of this manuscript for publication at this time.

However, while we are declining the current manuscript, we are in principle very interested in the work. We therefore would be open to a thoroughly revised manuscript that would be treated as a new submission.

– In the area of validation: a) presentation of data for knockdowns/overexpression, b) augmented validation of Cerox-1 and miR0488 interaction using RNA-FISH approaches and a clearer understanding of the effects on other genes after siRNA knockdowns.

– In the area of experimental design: a need to provide an explanation about the choice of HEK293T cell line for the comparative analysis in human in place of a line that was more comparative to the mouse line and clearer explanation of the analysis approaches used (see details in appended section.

In addition to these general comments there are numerous corrections and clarifications that would be needed to assist readers in understanding the data and conclusions made in the manuscript. These are also appended to this decision letter for your referral.

We thank you for your submission and hope that these comments may prove to be of some assistance.

*Reviewer #1:*

This manuscript describes work on a nc-RNA, Cerox-1 and its suggested role as a miRNA sponge for at least one post-transcriptionally regulating miRNA, miR488-3p. The modulating role of this miRNA is hypothesized to focus on levels of expression of the mitochondrial complex I subunit transcripts. The report is clearly written and timely since an model of coordinating the expression nuclear and mitochondrial genes needed for electron transport proteins is undeveloped. The proposed function of the nc RNA Cerox-1 as a linchpin in this regulation provides a contribution to both the ncRNA and mitochondrial fields of study.

However, there are three areas of consideration for improvement to aid in readers' understanding of the data and conclusions. First, consider transferring much of the information provided in the figure legends into the text of the manuscript since much of the legend information is needed for the continuation of the story being described in the text. Second, many of the figures display contain relatively small changes in enrichment or increased/decreased activities (considering the error bars for each experiment) after increased or decreased levels of expression of Cerox-1 or miR-488 (Figure 2D, Figures 3A.C, especially Figure 5A, D, Figure 6 A, B, D, F). While some of these are measured to be modestly significant other have no indication as to the statistical significance. It would be helpful to provide some general guidance to the readers as to the fact that which individual experiments may not be seen as having a compelling response, it is the overall trend that provides a convincing picture. The third areas concerns the interpretation of some of the arguments provided that rest primarily on correlations rather than definitive cause and effect evidence. An example of this is seen in the beginning of the manuscript arguing why CEROX-1 plays an important organismic role. Each of the points raised while noteworthy are either not surprising (conservation of promoter sequences) or observational or correlative. It would be helpful to readers for the authors to emphasize that some of the conclusions made are built on observations that are more consistent with the general conclusion rather than demonstratively evidential.

In summary the story and data presented in this manuscript will be illuminating to several different disciplines and raises provides additional evidence that ncRNAs are biologically important.

*Reviewer #2:*

In this study the authors identify and characterize a conserved cytoplasmic lncRNA, *Cerox1*, that regulates mitochondrial biology (levels of mitochondrial complex I subunit). This is proposed to occur by binding to microRNA-488-3p.

My major concerns with study can be distilled into the following three issues:

1) RNA-FISH validation of *Cerox1* localization and regulation.

It is important to perform RNA-FISH to validate the subcellular localization of *Cerox1* with respect to mitochondria. Although nuclear cytoplasmic fractionation is performed there is a lot of missing biology in the cytoplas with respect to where *Cerox1* is localized relative or within mitochondria. Finally it is important to perform co-fish of one or some of 13 genes from the mitichondrial genome that are upregulated between 1.4 fold and 3 fold by RNA sequencing (even though there were more modest increase above that for protein levels).

RNA-FISH would further address a missing aspects of this study: Does the lncRNA localize to the mitochondria or are these effects indirect (particularly with respect to the mitochondrial genome encoded regulatory events)? It is important to clarify how this possible form of regulation occurs. Whether by a cytoplasmic lncRNA localizing to mitochondria and regulating gene-expression changes, colocalizing with the regulatory protein products? Or is it via an indirect mechanism in a more diffuse cytoplasmic localization?

Does *Cerox1* co-localize with microRNA-488-3p? This would perhaps be more compelling evidence for observed effects being mediated by *Cerox1* binding microRNA-488-3p directly and or what fraction of microRNA-488-3p is sequestered.

2) Gain and Loss-of function studies.

The authors mention several experiments of increasing and decreasing *Cerox1* expression and monitoring several aspects (e.g., gene-expression and OXPHOS/Mito physiology). However, the primary data or amount of depletion of over-expression that was observed is not presented in any of the figures, the quantifications nor variances are mentioned in the text. I further looked for this information in the Materials and methods, Supplementary file 2 and related manuscript file. Considering the authors prior guidelines for lncRNA gain- and loss-of function strategies (Basset et al.) it is very important to include more clarification, primary data and further validation strategies described below.

A) For depletion studies the authors performed shRNA mediated depletions that have could have off target effects. It is important to test how much each shRNA depletes *Cerox1* and report this primary data and variances there in. Considering the off target possibilities with shRNAs (that cannot be computationally accounted for) it is important to have a secondary loss-of function strategy. Specifically, for the OXPHOS studies and gene-expression profiling. Perhaps most simply using CRISPR-I methodologies or CRISPR depletion studies of genetically defined clones.

B) Both shRNA and secondary depletion studies (e.g., crispr-I) should be validated by RNA-FISH to determine if specific sub-populations (as suggested above for the endogenous subcellular localization of *Cerox1*) are being depleted (e.g., near/in mitochondria) to further determine how discern potential direct or indirect regulation by *Cerox1*. Finally, this not only provides a more accurate and relevant quantification of depletion studies but also what fraction of cells demonstrated depletion.

C) Similar to (A and B) the gain-of function studies are not well described or effect sizes nor variances readily reported. This should be included as primary data in all relevant figures. The full-length cloning was described in the Materials and methods, however how much was over-expressed and or the nuclear cytoplasmic ratios of the subsequent transfection etc are not mentioned. It would also be important to report by RNA-FISH the subcellular localization of the combined in these over-expression studies. Finally, it is important to perform RNA-FISH on overexpression studies to determine what fraction of cells were exposed to over-expression conditions.

D) The effect sizes in gene-expression changes are overall very small and also with physiological experiments, albeit significant in most cases. For example the genes of focus are changing from 1.4 fold to 3 fold. Similar effect sizes are observed for several physiological measurement. For example the Complex IV component is the most increased by 1.6 fold. At most 2-3 fold changes are observed in respirations studies. Overall, the claims and conclusions in regulation, mechanism and physiology are all based on small cell-based effects; a concern when studying lncRNAs.

A loss or gain of function mouse model would mitigate perhaps more large-scale defects from the observed small-scale, cell based, perturbations. A mouse model demonstrating the organismal scale effects would be much more compelling to broad readership of *eLife*.

3) The mechanism suggested via microRNA-488-3p is preliminary and even smaller effect sizes. This possibility would be better validated by colocalization of microRNA-488-3p and *Cerox1* by RNA-FISH to determine if this is stoichiometrically possible

For example (PMID: 27871486):

"These results provide quantitative insights into the stoichiometric relationship between miRNAs and target abundance, target-site spacing, and affinity requirements for ceRNA-mediated gene regulation, and the unusual circumstances in which ceRNA-mediated gene regulation might be observed."

Overall this again points to understanding if these small effects are relevant on an organismal scale.

*Reviewer #3:*

The manuscript under review, "The long non-coding RNA *Cerox1* is a post-transcriptional regulator of mitochondrial complex I catalytic activity", submitted by T.M. Sirey et al., aims at elucidating potential roles of the lncRNA *Cerox1* in the post-transcriptional gene regulatory network of a number of mitochondrial complex I transcripts, by exerting a miRNA decoy activity against miR-488-3p.

In my opinion the study has a clear logic behind, and the manuscript sections – i.e. the order in which results are presented – are well organized. However, I would like to point out that I found some minor inconsistencies between what is reported in the text and the figures' captions. I will provide more details in the appropriate section below ("minor comments"), this is just a general comment to say that some parts of the manuscript may need to be explained in a clearer way.

As regards the experimental design, I only have two comments.

1) The authors focus on miR-488-3p, since it's the one, among the 4 selected miRNAs, that upon overexpression has the greatest effect in decreasing *Cerox1* transcript levels. I also noticed, even if it's not mentioned in the manuscript, that it's the miRNA with the highest number of predicted MREs in the 12 complex I transcripts (6 out of 12; Figure 5C). I understand the authors' interest in this specific miRNA, but I wonder whether they have planned to perform in the future further analyses at least with miR-370-3p, which is the one (after miR-488-3p) with the largest effect on *Cerox1* downregulation.

2) As concerns the analyses presented in Figure 7, I would have expected the authors to use a human cell line analogous to the one used for mouse, or at least derived from the nerve tissue, to be consistent in terms of comparative study. In more than one occasion, the authors point out the higher expression of *Cerox1* in both human and mouse adult brain with respect to other tissues. Besides, in the discussion they focus on the possible implications of mitochondrial dysfunction in neurological diseases like Parkinson's and Alzheimer's. I think the authors should provide an explanation about the choice of HEK293T cell line for the comparative analysis in human.

[Editors’ note: what now follows is the decision letter after the authors submitted for further consideration.]

Thank you for submitting your article "The long non-coding RNA *Cerox1* is a post transcriptional regulator of mitochondrial complex I catalytic activity" for consideration by *eLife*. Your article has been reviewed by two peer reviewers, and the evaluation has been overseen by a Reviewing Editor and Detlef Weigel as the Senior Editor. The reviewers have opted to remain anonymous. Reviewer #1 has seen the manuscript before, while reviewer #2 has not.

The reviewers have discussed the reviews with one another and the Reviewing Editor has drafted this decision to help you prepare a revised submission.

Summary:

This is a timely study, as there are still few examples of how lncRNAs act. In addition, linking lncRNAs to mitochondrial function makes this particular example especially interesting.

Essential revisions:

Please repeat the overexpression experiments and improve the protein blots, to demonstrate more clearly changes in the steady-state levels of complex I subunits.

It is not ideal that only one out of 6 shRNA constructs affected *Cerox1* levels, and even then only did so relatively modestly. Please screen an additional set of shRNA constructs. If none can be found with more substantial effects, you need to be more careful with your claim of reciprocal effects relative to overexpression.

*Reviewer #1:*

The authors have gone to great extents to better explain their results, add required data and validation by RNA FISH. Based on these explanations, clarifications and additions the authors have addressed my major concerns of effect size of knockdowns (the RNA FISH strongly reinforces cytoplasmic localization and shRNA efficiency - my biggest concern).

I only suggest that the authors include some of their key explanations such as more indirect models etc.

*Reviewer #2:*

The manuscript by Sirey et al. reports upon regulation of the levels of the subset of the mitochondrial complex I subunits by cytoplasmic long noncoding RNA (lncRNA), *Cerox1*. As this is the first report showing involvement of lncRNA in regulation of mitochondrial OXPHOS function this is a very timely study that would be of interest to a broader audience. The concept of the study is very interesting and the design of the experiment is logical and easy to follow, but in its current form, the presented evidence seems insufficient to prove the hypothesis put forward.

One of the most important conclusions in this study is a direct dependence of expression of the subset of complex I transcripts on *Cerox1* function, therefore a very careful analysis of this observation is needed before moving to further characterization (involvement of microRNAs) and conclusions. As the phenotypic effects are very weak, it is important for the reader be convinced that there is indeed significant evidence substantiating authors' claim.

Below are specific comments with suggestions for additional experiments:

Firstly, the authors claim that they provide evidence from reciprocal experiments (downregulation and overexpression of *Cerox1*) that *Cerox1* modulates expression of a subsets of complex I subunits, however, evidence coming from downregulation experiments is weak:

– Only one shRNA, moderately downregulating *Cerox1* expression (by 50%), is used. It is standard practice to use at least two shRNA to convincingly show that the effect is genuine and to avoid off-target problem.

– The authors observe a very mild effect (in any, as the p values are not shown, Figure 2—figure supplement 2E) of shRNA knockdown on transcripts of complex I subunits, but the effect on oxygen consumption is very pronounced. It is a quite puzzling observation. Normally, it is difficult to observe any effect on OXPHOS function, unless the mitochondrial complexes (transcripts and steady-state levels of the proteins) are substantially affected. What is even more puzzling is the fact that Seahorse analysis are done 12h after shRNA transfection. The steady state-levels of the mitochondrial complexes are high and their half-lives long. To my knowledge, there are no studies showing a dramatic effect on oxygen consumption after such a short time that would depend on the modulation of the levels of OXPHOS transcripts. In fact, it is a routine procedure in case of transfection with RNAi that would affect mitochondrial gene expression, to measure the levels and function of the OXPHOS complexes at least 3 or 6 days after RNAi treatment, when the effect starts being visible (for example Antonicka et al. EMBO Reports, 2017 doi: 10.15252/embr.201643391 or Richter et al. EMBO, DOI 10.1038/emboj.2010.14)

– Since there is such a dramatic effect on O2 consumption and almost no effect on transcripts level, to show a direct correlation and exclude a possibility of indirect effect, it is important to measure (for example by WB or quantitative mass spectrometry) the steady state levels of the complex I subunits after shRNA treatment.

Apart from the knockdown experiments, further validations are also needed to show a direct correlation between effect of *Cerox1* overexpression on complex I transcripts and increased OXPHOS performance:

– The qPCR validation of the expression of the OXPHOS subunits indicated by microarray shows rather weak correlation (although p values are needed to properly asses that). One could also argue that in the same time when complex I subunits are upregulated, components of complex IV are significantly downregulated (Figure 2—figure supplement 2D). However, it is activity of complex IV that is most significantly upregulated (Figure 3A).

– Importantly the western blot results presented in Figure 2D do not show convincingly upregulation of the investigated proteins and the quantifications presented are not reflected in shown WB blots. The quality of the part of the blot showing overexpression is poor (uneven loading -tubulin signal and diffused bands that are difficult to quantify). It would recommended to present additional blots with consistent results. Also, using other antibodies against components of complex I would be helpful (probably in Hek cell model as not many are validated in murine cells).

– One of the surprising results that has not been further investigated is a remarkable effect of *Cerox1* overexpression on cell growth. Why would a slight upregulation of OXPHOS components lead to dramatic growth retardation? There are many studies showing that boosting OXPHOS is beneficial for the cells. Considering that microarray results shown upregulation of more than 200 transcript, the effect on cell growth can be caused by another pathway, which in the same time can indirectly affect mitochondrial performance, hence upregulation of the OXPHOS function (not necessarily directly dependent on OXPHOS transcript elevation).

---

## [Author Response]

[Editors’ note: the author responses to the first round of peer review follow.]

While the reviewers found the subject matter of your manuscript is of interest and timely, they also found several issues in the manuscript that lowered their enthusiasm for the conclusions reached in the manuscript. The reviewers find the manuscript problematic in several places focused on experimental design, validation and small effect phenotypic changes supporting conclusions from targeted genetic perturbation experiments. A summary of the reviews are enclosed at the end of this letter for your review. Thus, unfortunately we are unable to recommend the acceptance of this manuscript for publication at this time.

We address concerns regarding experimental design (HEK293T cells) and validation (data presentation and RNA-FISH) in our comments directly below. With regard to the third comment, we disagree with the view that the phenotypic changes are “of small effect”. The changes we observed in transcript (mRNA) levels are, indeed, ~20-50%. Yet the greatest changes occur for other features and these are the most proximal to cellular phenotype. A) Complex I protein levels double (Figure 2D) which will have a large effect because of these proteins’ high cellular abundance and their half-lives being exceedingly long (>days). B) Complex I or IV activities change by ~10-50% which is the level of change observed in pathological states (Parkinson’s or Alzheimer’s diseases). These are substantial increases in mitochondrial complex activities. (C) Increased *Cerox1* levels nearly halved ROS production (Figure 4A) and cell cycle activity (without changing cell cycle proportions; Figure 2—figure supplement 1H,I,J). C) It also can protect cells from the effect of rotenone (~40%) and sodium azide (~80%), and from oxidative stress (Figure 4C). None of these effects i) could be considered small, or ii) was previously suspected to be mediated by a lncRNA or a miRNA. These points are now given greater emphasis in the revised manuscript.

However, while we are declining the current manuscript, we are in principle very interested in the work. We therefore would be open to a thoroughly revised manuscript that would be treated as a new submission.– In the area of validation: a) presentation of data for knockdowns/overexpression, b) augmented validation of Cerox-1 and miR0488 interaction using RNA-FISH approaches and a clearer understanding of the effects on other genes after siRNA knockdowns.

a) As requested, we have made suggested changes throughout the manuscript to improve the presentation of data. These include relocation of numerical results (e.g. p-values) into the text to improve the continuity of the narrative, and some further indications of statistical significance. b) As requested we have performed dual RNA-FISH (new Figure 5F) but note that it is not technically feasible to use this approach for demonstrating colocalisation of *Cerox1* with miR-488-3p. This is because this miRNA necessarily competes for both *Cerox1* and the miR-488-3p-complementary FISH probes owing to its limited length. We consider most definitive our previous observation of a direct physical interaction between miR-488-3p with *Cerox1* and each of 4-10 complex I subunit’s mRNAs (Figure 6G). c) The clearest effect of knocking down *Cerox1* is on cellular basal, ATP-linked and uncoupled respiration (Seahorse data: Figure 3D). As we discuss above, these effects are more proximal indicators of cellular phenotypes than transcript level changes. The shRNA that we used for these experiments is clearly validated by the reciprocal effects we observed in each over-expression assay.

– In the area of experimental design: a need to provide an explanation about the choice of HEK293T cell line for the comparative analysis in human in place of a line that was more comparative to the mouse line and clearer explanation of the analysis approaches used (see details in appended section.

As we explain below, we had 3 reasons for choosing HEK293T cells for this experiment. 1) They are of neural crest ectodermal origin (Lin et al., 2014) and therefore have a number of neural cell line characteristics including their expression of neuronal markers (Shaw et al., 2002). 2) They are in common use as a cell culture model for neurodegenerative diseases such as Parkinson’s disease (Falkenburger et al., 2016; Schlachetzki et al., 2013). 3) HEK293T cells are a widely used cell line to interrogate human mitochondrial biochemistry, and exhibit complex I dependent respiration (Kim et al., 2014). These points are now included in the revised manuscript (Tissue culture and flow cytometry Section, Materials and methods). Further details of our analytical approaches have now been provided in the Materials and methods and elsewhere.

Reviewer #1:

This manuscript describes work on a nc-RNA, Cerox-1 and its suggested role as a miRNA sponge for at least one post-transcriptionally regulating miRNA, miR488-3p. The modulating role of this miRNA is hypothesized to focus on levels of expression of the mitochondrial complex I subunit transcripts. The report is clearly written and timely since an model of coordinating the expression nuclear and mitochondrial genes needed for electron transport proteins is undeveloped. The proposed function of the nc RNA Cerox-1 as a linchpin in this regulation provides a contribution to both the ncRNA and mitochondrial fields of study.However, there are three areas of consideration for improvement to aid in readers' understanding of the data and conclusions. First, consider transferring much of the information provided in the figure legends into the text of the manuscript since much of the legend information is needed for the continuation of the story being described in the text.

In response, in several places we now relocate into the Results p-values and measurements that are critical for providing continuity.

Second, many of the figures display contain relatively small changes in enrichment or increased/decreased activities (considering the error bars for each experiment) after increased or decreased levels of expression of Cerox-1 or miR-488 (Figure 2D, Figures 3A, C, especially Figure 5A, D, Figure 6 A, B, D, F). While some of these are measured to be modestly significant other have no indication as to the statistical significance. It would be helpful to provide some general guidance to the readers as to the fact that which individual experiments may not be seen as having a compelling response, it is the overall trend that provides a convincing picture.

Our results show that:

• NDUFS1 and NDUFS3 protein levels double (1.9- and 2.1-fold) – Figure 2D;

• *Cerox1* overexpression increases Complex I or IV activities by 20% or 50% whilst knockdown reduces Complex I and IV activities by 11% or 19% – Figure 3A,C;

• mRNA expression increases by 1.3-1.6-fold –Figure 5A;

• *Cerox1* overexpression increases Complex I or IV activities by 30% and 17% –Figure 5D; and,

• there are changes in mRNA levels (~20% [Figure 6A], 50% [Figure 6B]), in luciferase activity (~15% [Figure 6D]), and in complex I enzyme activity (~30% [Figure 6F]).

The reviewer describes these changes as small relative to the error bars, although – where indicated – these are all statistically significant. The reviewer, we believe wrongly, conflates mRNA expression change with cellular outcome when, instead, changes in absolute level of protein and enzyme activity are the most proximal and thus important indicators of outcome. Our evidence for this is two-fold:

A) Changes that are most biologically relevant are to the protein and enzyme activity levels, and these changes greatly exceed the mRNA changes (as we described in our original submission). Complex I protein half-lives are exceedingly long, > several days (see new Figure 2—figure supplement 2G) (Dorrbaum et al., 2018; Mathieson et al., 2018; Schwanhäusser et al., 2011), their transcripts are stable (Friedel et al., 2009; Schwanhäusser et al., 2011; Sharova et al., 2009; Tani et al., 2012) (see Figure 2—figure supplement 1H) and their levels among the highest for many cell types (Schwanhäusser et al., 2011) implying that *Cerox1*-mediated changes to cellular metabolism are both long-lasting and more profound than immediately apparent from the observed mRNA changes.

B) The level of change in mitochondrial complex activity we observed (~10%-50%) is equivalent to the decreases in activity observed in Parkinson’s and Alzheimer’s diseases; complex I: 30% (Keeney et al., 2006; Schapira et al., 1990); complex IV: 40% (Canevari et al., 1999) or to the increase in activity of complex I in dopaminergic neurons that protects these cells against a complex I toxin (MPTP) in mouse N2A cells (Alvarez-Fischer et al., 2011). When considered against the observation that a human body turns over more than its weight in ATP daily (Brooks, 1998; Brooks et al., 2004; Gaesser and Brooks, 1975; Pickart and Jencks, 1984), such increases in mitochondrial complex activities are both substantial and most likely consequential.

To further address the reviewer’s point, we investigated the cellular outcomes of *CEROX1* expression change by determining the oxygen consumption change in human HEK293T cells using a Seahorse XFe24 Analyzer. These new data (now provided in the revised manuscript as Figure 7E) shows a 30%-35% increase (Figure 7E) in cellular respiration. Again, the cumulative and large change to respiration is expected to have a substantial knock-on effect on cellular function. Overall, our data could explain the large (~40%) reduction in cell cycle activity when *Cerox1* is overexpressed.

The manuscript has been modified to include long half-life data (subsection “*Cerox1* expression modulates levels of oxidative phosphorylation transcripts” paragraph three and “*Cerox1* can regulate mitochondrial OXPHOS enzymatic activity” paragraph two; Figure 2—figure supplement 1G and H) and detail on the Seahorse experiments (subsection “*Cerox1* can regulate mitochondrial OXPHOS enzymatic activity” paragraph two, Figure 3B,D, Figure 7E, subsection “Oxidative phosphorylation enzyme assays and oxygen consumption” paragraph four). In a few instances (Figure 6A and 6B) we now include indicators of statistical significance that were missing previously.

The third areas concerns the interpretation of some of the arguments provided that rest primarily on correlations rather than definitive cause and effect evidence. An example of this is seen in the beginning of the manuscript arguing why CEROX-1 plays an important organismic role. Each of the points raised while noteworthy are either not surprising (conservation of promoter sequences) or observational or correlative. It would be helpful to readers for the authors to emphasize that some of the conclusions made are built on observations that are more consistent with the general conclusion rather than demonstratively evidential.

Most evidence in the submission relates to definitive cause-and-effect: specifically, the reciprocal consequences on cellular mRNA and protein abundance, enzyme activity, cell viability and cell cycle activity, following the increase and decrease of *Cerox1* RNA levels. We consider the observations that miR-488-3p directly binds *Cerox1* (and mRNAs of mitochondrial complex subunits) and the reciprocality of effect of human *CEROX1* in mouse cells, and vice versa (Figure 7F), to be particularly strong findings. The example the reviewer cites from “the beginning of the manuscript” is clearly stated (Results paragraph two) as supporting an organismal role rather than as definitively playing this role. In redrafting the manuscript we have taken extra care to separate correlative from direct evidence for our interpretation.

We also wish to draw the reviewers’ attention to additional information that has been published as a bioRxiv preprint since our initial submission that further supports the organismal role of *Cerox1*. In a mouse model of haematopoietic lineage differentiation depletion of lncRNAs highly expressed in haematopoietic stem cells leads to a deficit in B-lymphocyte differentiation (*Cerox1,* in this instance called lnc6689; Delás et al., 2019). We have updated the manuscript with this preprint citation.

In summary the story and data presented in this manuscript will be illuminating to several different disciplines and raises provides additional evidence that ncRNAs are biologically important.

We are pleased that the reviewer believes that the manuscript is highly relevant to several disciplines and provides evidence for an unanticipated role for a lncRNA in regulating mitochondrial function.

Reviewer #2:

In this study the authors identify and characterize a conserved cytoplasmic lncRNA, Cerox1, that regulates mitochondrial biology (levels of mitochondrial complex I subunit). This is proposed to occur by binding to microRNA-488-3p.My major concerns with study can be distilled into the following three issues:1) RNA-FISH validation of Cerox1 localization and regulation.It is important to perform RNA-FISH to validate the subcellular localization of Cerox1 with respect to mitochondria. Although nuclear cytoplasmic fractionation is performed there is a lot of missing biology in the cytoplas with respect to where Cerox1 is localized relative or within mitochondria.

The reviewer requested additional evidence of *Cerox1* localisation to the cytoplasm. Please note that our proposed mechanism does not require *Cerox1* to be proximal to mitochondria, and also that at no point in the manuscript did we claim, or mean to imply, that *Cerox1* is localised within the mitochondrion. As requested, we used RNA-FISH to validate our fractionation results that *Cerox1* is cytoplasmic in N2A cells (new Figure 1F).

*Cerox1* will be in close proximity to the mitochondrial network because these are both dispersed widely throughout the cytoplasm of N2A cells (*Cerox1*: Figure 1F; mitochondrial network: e.g. (Geng et al., 2018; Yeo et al., 2015)). Moreover, each is found within or adjacent to ribosomes: mouse *Cerox1* is associated with the ribosome (our data: Figure 1—figure supplement 1I), as are 94% (17/18) of human *CEROX1* transcript reads (van Heesch et al., 2014); ribosomes present on the rough-ER also often lie adjacent to mitochondria (Eskelinen, 2008; Wu et al., 2017).

Finally it is important to perform co-fish of one or some of 13 genes from the mitichondrial genome that are upregulated between 1.4 fold and 3 fold by RNA sequencing (even though there were more modest increase above that for protein levels).

In our original submission we were clear that “The 15 ETC transcripts that show statistically significant differential expression after *Cerox1* overexpression are nuclear [not mitochondrial genome] encoded (Figure 2B,C)”.

The reviewer is seeking experimental evidence for the co-localisation of *Cerox1*, miR-488-3p, and multiple mitochondrial complex I mRNAs, presumably at the rough ER – mitochondria interface. Nevertheless, colocalisation is not a requirement for the model we propose. Mitochondrial complex I mRNAs are abundant, have long half-lives, and – like *Cerox1* – are diffusible in the cytosol. The model does not require *Cerox1* and mitochondrial subunit mRNAs to physically interact. It only requires: (i) that miR-488-3p can independently bind *Cerox1* or a mitochondrial complex I mRNA, and (ii) that each such complex, within the RISC, results in destabilisation of the transcript. This model does not require *Cerox1* to be colocalised with any mitochondrial complex I mRNA in the cell.

The only molecular colocalisation required for the model is the physical association of miR488-3p with *Cerox1* and, separately, with each of the mitochondrial complex I mRNAs. It is this direct, physical interaction that we previously demonstrated (Figure 6G).

RNA-FISH would further address a missing aspects of this study: Does the lncRNA localize to the mitochondria or are these effects indirect (particularly with respect to the mitochondrial genome encoded regulatory events)? It is important to clarify how this possible form of regulation occurs. Whether by a cytoplasmic lncRNA localizing to mitochondria and regulating gene-expression changes, colocalizing with the regulatory protein products? Or is it via an indirect mechanism in a more diffuse cytoplasmic localization?

In response to the reviewer’s comment, we used dual RNA-FISH to show that both *Cerox1* and miR-488-3p are localised in the cytoplasm (Figure 5F [in normal, overexpression and knockdown N2A cells]). The two ncRNAs remain in the cytoplasm in the knockdown and overexpression conditions (see Author response image 1 – images are projections from Z-stacks). Human *CEROX1* transcript reads are found predominantly (94%) localised to ribosomes (van Heesch et al., 2014), and the destabilisation by microRNAs of mRNAs as they are being translated is also localised to ribosomes (Tat et al., 2016). Together, these observations imply that *Cerox1* and mitochondrial protein mRNAs are targets of miR-488-3p on ribosomes within the rough ER that forms a network around mitochondria (Wu et al., 2017). Indeed, the study’s data all support a model involving (as the reviewer described) “an indirect mechanism in a more diffuse cytoplasmic localization” that, consequently, does not expect colocalization of *Cerox1*/complex I mRNAs. We now include these observations in the revised manuscript (subsection “Increased OXPHOS enzymatic activity is dependent upon miRNA binding to *Cerox1*” paragraph three).

Does Cerox1 co-localize with microRNA-488-3p? This would perhaps be more compelling evidence for observed effects being mediated by Cerox1 binding microRNA-488-3p directly and or what fraction of microRNA-488-3p is sequestered.

Our previous evidence for direct physical interaction (Figure 6G) provides compelling evidence for these effects being mediated directly. The dual RNA-FISH images (new Figure 5F) show that both *Cerox1* and miRNA-488-3p are localised within the cytoplasm. RNA-FISH is not technically able to demonstrate colocalisation in cells because miR-488-3p will compete for both its *Cerox1* MRE and the short FISH probes complementary to miR-488-3p. Nevertheless, using pull-down assays we previously demonstrated direct physical interaction between miR488-3p and *Cerox1* (and indeed with many mitochondrial complex subunit mRNAs) (Figure 6G). This direct physical interaction explains the effect of miR-488-3p overexpression and inhibition on *Cerox1* levels (Figure 6A,B), and the loss of activity, in a luciferase assay, of the *Cerox1* transcript mutated in the miR-488-3p MRE (Figure 6D).

2) Gain and Loss-of function studies.The authors mention several experiments of increasing and decreasing Cerox1 expression and monitoring several aspects (e.g., gene-expression and OXPHOS/Mito physiology). However, the primary data or amount of depletion of over-expression that was observed is not presented in any of the figures, the quantifications nor variances are mentioned in the text. I further looked for this information in the Materials and methods, Supplementary file 2 and related manuscript file. Considering the authors prior guidelines for lncRNA gain- and loss-of function strategies (Basset et al.) it is very important to include more clarification, primary data and further validation strategies described below.

We apologise that some required data was not present in the original submission. In all experiments, transfections were optimised such that *Cerox1* overexpression was reproducibly 6-to-8 fold increased, or 50-60% knocked down. Of the 7 Figure panels showing gene expression qPCR data (Figures 2C, 5A, 5C, 6A, 6B, 6E, 6G), *Cerox1* expression levels were included for 4 (Figures 5A, 6A, 6B and 6G) and are now shown, in the revised manuscript, for the remaining 3 (Figures 2C, 5C, 6E). For all other assays requiring *Cerox1* overexpression or depletion its level was not measurable owing to conflicting sample preparation protocols, but these, again, were performed using the identical optimised experimental transfection conditions.

A) For depletion studies the authors performed shRNA mediated depletions that have could have off target effects. It is important to test how much each shRNA depletes Cerox1 and report this primary data and variances there in. Considering the off target possibilities with shRNAs (that cannot be computationally accounted for) it is important to have a secondary loss-of function strategy. Specifically, for the OXPHOS studies and gene-expression profiling. Perhaps most simply using CRISPR-I methodologies or CRISPR depletion studies of genetically defined clones.

We tested six shRNA constructs designed using a combination of algorithms from the Whitehead Institute and Invitrogen and locations predicted by both tools were chosen for shRNA vector construction (Materials and methods). Only one shRNA depleted *Cerox1* by greater than 50% (now provided as Figure 2—figure supplement 1A). We are aware that shRNAs can have off-target effects, yet our full set of observations cannot be explained by these effects: this shRNA shows a reciprocal phenotype to that observed in the *Cerox1* overexpression assay, at the level of: a) gene expression, b) mitochondrial complex I and complex IV enzyme activity and c) overall mitochondrial oxygen consumption.

We had substantial reservations about attempting CRISPR-A and -I at the *Cerox1* locus for 3 reasons: 1) the close proximity and shared bi-directional promoter with the protein coding gene *Sox8* (Figure 1A, Figure 1—figure supplement 1A), 2) the presence of a large CpG island overlapping the transcriptional start sites of both *Cerox1* and *Sox8*, and 3) reports that multiple genes are up/down regulated from bidirectional promoter regions leading to false negative and false positive results (Rosenfeld et al., 2011; Sanson et al., 2018) and that bidirectional promoters are a known challenge for the ‘crisprability’ of transcripts (Goyal et al., 2017).

Nevertheless, to test the utility of CRISPR at the *Cerox1* locus we tiled 6 guide RNAs across the shared bidirectional *Cerox1/Sox8* promoter within the recommended windows for CRISPR-A and CRISPR-I (Gilbert et al., 2014) and assayed for changes in gene expression of the adjacent genes *Cerox1, Sox8* and Lmf1. Neither CRISPR-A nor CRISPR-I had any effect on the expression of Lmf1. However, CRISPR-A significantly upregulated *Sox8* expression (by, on average, 5.8 fold and 13.6 fold in two separate experiments; new Figure 2—figure supplement 1B), whilst having only a small and variable effect on *Cerox1* expression (average 1.1 fold and 0.8 fold; Figure 2—figure supplement 1B). Again, CRISPR-I had a small variable effect on the expression of *Cerox1* (on average 1.3 fold and 0.7 fold; Figure 2—figure supplement 1C), but also had an effect on *Sox8* expression (Figure 2—figure supplement 1C). We concluded that current CRISPRi/a approaches cannot be used to specifically target *Cerox1* expression independent of *Sox8* expression.

B) Both shRNA and secondary depletion studies (e.g., crispr-I) should be validated by RNA-FISH to determine if specific sub-populations (as suggested above for the endogenous subcellular localization of Cerox1) are being depleted (e.g., near/in mitochondria) to further determine how discern potential direct or indirect regulation by Cerox1. Finally, this not only provides a more accurate and relevant quantification of depletion studies but also what fraction of cells demonstrated depletion.

As part of our procedure for optimising transfection, cells were cotransfected with a GFP reporter and either the overexpression construct or the shRNA vector. After 24h transfection efficiencies were assessed first by microscopy, and then by FACS. On average transfection efficiency (i.e. GFP signal) was between 70%-80% for the overexpression construct (n=6) and 55%-75% for the shRNA construct (n=6) – see Author response image 2. As analysis by FACS was not always convenient after initial transfection optimisation, transfection efficiency was assessed by microscopy using the GFP positive transfection control.

**Author response image 2. respfig2:** 

C) Similar to (A and B) the gain-of function studies are not well described or effect sizes nor variances readily reported. This should be included as primary data in all relevant figures. The full-length cloning was described in the Materials and methods, however how much was over-expressed and or the nuclear cytoplasmic ratios of the subsequent transfection etc are not mentioned. It would also be important to report by RNA-FISH the subcellular localization of the combined in these over-expression studies.

For gain-of-function (i.e. overexpression) studies, cells were transfected with a construct overexpressing *Cerox1* and harvested after 48 hrs (as described previously in the Materials and methods). We now provide further details of 5’ and 3’ RACE, and cloning in the Materials and methods. Again, we apologise for insufficient data in the submission: (a) Data regarding the optimised overexpression and depletion of *Cerox1* gene expression levels (with neighbouring protein coding genes) are presented in Figure 2—figure supplement 1D; and, (b) Transfections were optimised so that *Cerox1* overexpression was consistently 6-8 fold increased, or 50%-60% knocked down. These experimental conditions were consistent for all experiments. As we described above, all 7 gene expression qPCR Figure panels (2C, 5A, 5C, 6A, 6B, 6E, 6G) now show *Cerox1* expression level and standard error of the mean. RNA-FISH (Figure 5F) shows that *Cerox1* overexpression occurs, as expected, within the cytoplasm. (We hope that we have understood the reviewer’s incomplete last sentence, above.)

Finally, it is important to perform RNA-FISH on overexpression studies to determine what fraction of cells were exposed to over-expression conditions.

RNA-FISH on *Cerox1* overexpression cells confirms its higher abundance, and its location, together with miR-488-3p, in the cytoplasm. Transfection efficiency in N2A cells was high (~70%-80%, see Author response image 2). Our experience is that assessment of transfection efficiency by FACS yields greater accuracy than RNA-FISH.

D) The effect sizes in gene-expression changes are overall very small and also with physiological experiments, albeit significant in most cases. For example the genes of focus are changing from 1.4 fold to 3 fold. Similar effect sizes are observed for several physiological measurement. For example the Complex IV component is the most increased by 1.6 fold. At most 2-3 fold changes are observed in respirations studies. Overall, the claims and conclusions in regulation, mechanism and physiology are all based on small cell-based effects; a concern when studying lncRNAs.A loss or gain of function mouse model would mitigate perhaps more large-scale defects from the observed small-scale, cell based, perturbations. A mouse model demonstrating the organismal scale effects would be much more compelling to broad readership of eLife.

We would like to reiterate our response to reviewer 1’s second point (see above) and include additional comments regarding the cellular respiration data. Briefly, it would be wrong, we maintain, for mRNA expression change to be conflated with cellular outcome. Changes in absolute level of protein and enzyme activity are the most important indicators of outcome because:

A) Changes that are most biologically relevant are to the protein and enzyme activity levels, and these changes greatly exceed the mRNA changes (as we described in our original submission). Complex I protein half-lives are exceedingly long, > several days (see new Figure 2—figure supplement 1G) (Dorrbaum et al., 2018; Mathieson et al., 2018; Schwanhäusser et al., 2011), their transcripts are stable (Friedel et al., 2009; Schwanhäusser et al., 2011; Sharova et al., 2009; Tani et al., 2012) (see new Figure 2—figure supplement 1H) and their levels among the highest for many cell types (Schwanhäusser et al., 2011) implying that *Cerox1*-mediated changes to cellular metabolism are both long-lasting and more profound than immediately apparent from the observed mRNA changes.

B) The level of change in mitochondrial complex activity we observed (~10%-50%) is equivalent to the decreases in activity observed in Parkinson’s and Alzheimer’s diseases; complex I: 30% (Keeney et al., 2006; Schapira et al., 1990); complex IV: 40% (Canevari et al., 1999) or to the increase in activity of complex I in dopaminergic neurons that protects these cells against a complex I toxin (MPTP) in mouse N2A cells (Alvarez-Fischer et al., 2011). When considered against the observation that a human body turns over more than its weight in ATP daily (Brooks, 1998; Brooks et al., 2004; Gaesser and Brooks, 1975; Pickart and Jencks, 1984), such increases in mitochondrial complex activities are both substantial and most likely consequential.

In addition, the oxygen consumption measurements we report in both N2A and HEK293T cells are not dissimilar in magnitude to those reported for the manipulation of protein coding genes or drug treatments that exhibit a respiratory phenotype in these cell lines. For instance, in N2A cells the silencing of the mitochondrial aspartate-glutamate carrier isoform AGC1, defects in which cause infant encephalopathy with delayed myelination, results in a ~44% decrease in basal respiration (Profilo et al., 2017), whilst treatment of N2A cells with 1,3,4-oxadiazole derivatives results in a ~27% and 29% decrease in basal respiration and ATP-linked respiration respectively (Tok et al., 2018). In HEK cells investigation of glucocorticoid receptor (GR) isoforms demonstrated that GR-γ increased basal respiration by ~17%, and ATP-linked respiration by ~29% (Morgan et al., 2016), whilst cells devoid of manganese superoxide dismutase (MnSOD) – an enzyme essential for protecting the cells from oxidative damage – exhibited a 59% decrease in basal respiration (Cramer-Morales et al., 2015). Knock out of acylglycerol kinase (AGK) which is mutated in Sengers Syndrome decreases basal respiration by 26% (Vukotic et al., 2017). We note that *Cerox1* overexpression in N2A cells results in respiratory changes that are larger (85% increase in basal respiration and 107% increase in ATP-linked respiration) than most previous reports. Consequently, our observed changes in respiration are comparable to, or exceed, those observed following the manipulation of respiratory relevant protein coding genes or the addition of drugs that affect mitochondrial oxygen consumption.

We are, indeed, making *Cerox1*-deficient mouse models (having first suffered from a collaborator’s design which unfortunately also knocked out expression of the neighbouring *Sox8* gene) and intend to describe, in a separate publication, in-depth anatomical, developmental, metabolic and behavioural characterizations, as we have done previously (Oliver et al., 2015). Our current report is timely and important because it presents molecular and cellular mechanistic insights into how interacting noncoding RNAs can co-ordinately regulate mitochondrial oxidative phosphorylation, as supported by reviewer 1 (“[it] will be illuminating to several different disciplines and … provides additional evidence that ncRNAs are biologically important”).

3) The mechanism suggested via microRNA-488-3p is preliminary and even smaller effect sizes.

We contest the view that the miR-488-3p mechanism is preliminary for 3 reasons: (I) Overexpression of miR-488-3p causes substantial depletion not just of *Cerox1* (Figure 5E) but also of a dozen complex I subunit mRNAs (Figure 6A). The depletion levels may not have impressed the reviewer because their proportional decrease is ~20%, yet these changes are remarkable because of the extreme (above median level) abundance of these mRNAs and, more importantly, their encoded proteins within the 1k-2k mitochondria in the cell occupying ~20% of its volume. (II) MRE mutation. Abrogation of the effects on RNA expression and on complex I enzymatic activity on overexpressing a mutant *Cerox1* transcript mutated in its miR-488-3p response element (Figure 6E, F). (III) Direct binding interaction measured between miR-4883p and *Cerox1* and numerous other complex I mRNAs (Figure 6G).

Furthermore, there is a large (43%) change in overall timing of cell division caused by *Cerox1* overexpression, and the enzymatic activities of complexes I and IV whose extremely high basal rates are increased yet further by 22% and 50%, respectively (Figure 3A). Our response to 2D (above) is again relevant here.

This possibility would be better validated by colocalization of microRNA-488-3p and Cerox1 by RNA-FISH to determine if this is stoichiometrically possible

This is the reviewer’s point 1) to which we responded: “RNA-FISH is not technically able to demonstrate colocalisation in cells because miR-488-3p will compete for both its *Cerox1* MRE and the short FISH probes complementary to miR-488-3p.”

This reviewer then quotes from Denzler et al., 2016; “Impact of MicroRNA Levels, Target-Site Complementarity, and Cooperativity on Competing Endogenous RNA-Regulated Gene Expression.”

For example (PMID: 27871486):"These results provide quantitative insights into the stoichiometric relationship between miRNAs and target abundance, target-site spacing, and affinity requirements for ceRNA-mediated gene regulation, and the unusual circumstances in which ceRNA-mediated gene regulation might be observed."

The findings of this Denzler et al., 2016 are not definitive, in particular because: “the conclusions drawn regarding the physiological relevance of ceRNA crosstalk by both Bosson et al., 2014, and Denzler et al., 2014; 2016, rely on assumptions that are at odds with experimental observations” and “despite the conclusions of others (Denzler et al., 2016), it remains plausible that ceRNA crosstalk occurs within a physiological range of gene expression but only for a subset of transcripts that are distinguished by their efficiency at recruiting and binding miRNAs.” Here we are quoting from our extensive, and well-received, review on this issue (Smillie et al., 2018) whose arguments are relevant to this controversy, but for brevity need not be repeated here.

Overall this again points to understanding if these small effects are relevant on an organismal scale.

We have provided evidence, using diverse methodologies, that *Cerox1*- and miR-488-3pmediated expression changes, of the order of 20% in RNA levels, have subsequent greater effects (~2-fold) on the levels of proteins that have long (~multiple day) half-lives and on the enzymatic activities of mitochondria (~1.6-2.1-fold, Figure 3B) that produce more than each person’s mass in ATP each day.

Reviewer #3:

The manuscript under review, "The long non-coding RNA Cerox1 is a post-transcriptional regulator of mitochondrial complex I catalytic activity", submitted by T.M. Sirey et al., aims at elucidating potential roles of the lncRNA Cerox1 in the post-transcriptional gene regulatory network of a number of mitochondrial complex I transcripts, by exerting a miRNA decoy activity against miR-488-3p.In my opinion the study has a clear logic behind, and the manuscript sections – i.e. the order in which results are presented – are well organized. However, I would like to point out that I found some minor inconsistencies between what is reported in the text and the figures' captions. I will provide more details in the appropriate section below ("minor comments"), this is just a general comment to say that some parts of the manuscript may need to be explained in a clearer way.

Thank you for your supportive comments. We have addressed these minor inconsistencies in the revised manuscript, as we describe in detail below.

As regards the experimental design, I only have two comments.1) The authors focus on miR-488-3p, since it's the one, among the 4 selected miRNAs, that upon overexpression has the greatest effect in decreasing Cerox1 transcript levels. I also noticed, even if it's not mentioned in the manuscript, that it's the miRNA with the highest number of predicted MREs in the 12 complex I transcripts (6 out of 12; Figure 5C). I understand the authors' interest in this specific miRNA, but I wonder whether they have planned to perform in the future further analyses at least with miR-370-3p, which is the one (after miR-488-3p) with the largest effect on Cerox1 downregulation.

At the time we assayed for the effect of overexpressing miR-370-3p in N2A cells. This experiment was not included in the submission because it provided no evidence for the direct repression of our biomarker transcripts, or indeed *Cerox1* whose level increased (see Author response image 3).

**Author response image 3. respfig3:** 

2) As concerns the analyses presented in Figure 7, I would have expected the authors to use a human cell line analogous to the one used for mouse, or at least derived from the nerve tissue, to be consistent in terms of comparative study. In more than one occasion, the authors point out the higher expression of Cerox1 in both human and mouse adult brain with respect to other tissues. Besides, in the discussion they focus on the possible implications of mitochondrial dysfunction in neurological diseases like Parkinson's and Alzheimer's. I think the authors should provide an explanation about the choice of HEK293T cell line for the comparative analysis in human.

We had three reasons for choosing HEK293T cells for this experiment. First, they are of neural crest ectodermal origin (Lin et al., 2014) and therefore have a number of neural cell line characteristics in that they express neuronal markers (Shaw et al., 2002). Second, they are in use as a cell culture model for neurodegenerative diseases such as Parkinson’s disease (Falkenburger et al., 2016; Schlachetzki et al., 2013). Third, HEK293T cells are a widely used cell line to interrogate human mitochondrial biochemistry, and exhibit complex I dependent respiration (Kim et al., 2014). We now include these points in the revised manuscript.

[Editors' note: the author responses to the re-review follow.]

Essential revisions:Please repeat the overexpression experiments and improve the protein blots, to demonstrate more clearly changes in the steady-state levels of complex I subunits.

These experiments have been repeated, and new and improved Western blot images have been provided as a new version of Figure 2D. The additional data has been added to the original data and are displayed as box plots. Protein levels of NDUFS1 and NDUFS3 are significantly elevated upon *Cerox1* overexpression (p<0.01).

It is not ideal that only one out of 6 shRNA constructs affected Cerox1 levels, and even then only did so relatively modestly. Please screen an additional set of shRNA constructs. If none can be found with more substantial effects, you need to be more careful with your claim of reciprocal effects relative to overexpression.

In response, six additional shRNAs were newly screened for their ability to knock-down *Cerox1*. None of these constructs were as effective in knocking down *Cerox1* as sh92 (50-60% knockdown Figure 2—figure supplement 1A), but one construct (sh1159) was the next best shRNA, decreasing *Cerox1* expression by 40%. Consequently, we examined the effect of sh1159 mediated knock down of *Cerox1* on i) the expression of complex I subunit transcripts, and ii) its effect on cellular oxygen consumption. i) *Cerox1* knock-down by sh1159 resulted in significant decreases in levels of Complex I subunit transcripts (*Ndufs6, Ndufab1, Ndufs3* and *Ndufs1*; new Figure 2—figure supplement 1F), noting that these transcripts are also significantly decreased in abundance when *Cerox1* was knocked down with sh92 (Figure 2—figure supplement 1E). ii) *Cerox1* knock down using sh1159 resulted in lower median basal respiration, ATP-linked respiration and maximum uncoupled respiration (see Author response image 4) although these results did not reach statistical significance (i.e. p>0.05). Nevertheless, results from new experiments i) and ii) are consistent with our previous data (acquired from *Cerox1* overexpression assays) in indicating that the action of sh92 is not due to off-target binding

**Author response image 4. respfig4:** 

Reviewer #1:

The authors have gone to great extents to better explain their results, add required data and validation by RNA FISH. Based on these explanations, clarifications and additions the authors have addressed my major concerns of effect size of knockdowns (the RNA FISH strongly reinforces cytoplasmic localization and shRNA efficiency – my biggest concern).

We would like to thank reviewer 1 for their helpful suggestions that brought further clarity to our results, and for suggesting the FISH experiments that clarified the cellular localisation of *Cerox1* and the efficiency of the shRNA.

I only suggest that the authors include some of their key explanations such as more indirect models etc.

The likelihood of indirect effects has now been added to the Discussion: “These genes’ transcripts will be both the targets of miR-488-3p decoying by *Cerox1* (Figure 6) and those whose upregulation is secondary to *Cerox1* (and miR-488-3p) mediated effects, for example relating to the observed changes in cellular metabolism and proliferation (Figure 2—figure supplement 1I, J, K).”

Reviewer #2:

The manuscript by Sirey et al. reports upon regulation of the levels of the subset of the mitochondrial complex I subunits by cytoplasmic long noncoding RNA (lncRNA), Cerox1. As this is the first report showing involvement of lncRNA in regulation of mitochondrial OXPHOS function this is a very timely study that would be of interest to a broader audience. The concept of the study is very interesting and the design of the experiment is logical and easy to follow, but in its current form, the presented evidence seems insufficient to prove the hypothesis put forward.

Thank you for these comments. The evidence for i) direct physical interactions between miR-488 and *Cerox1* (and complex I subunit transcripts) and ii) the abolition of downstream effects on complex I transcript levels and complex I activity upon targeted mutation of the miR-488 MRE in *Cerox1* (Figure 5C,D; Figure 6D, E, F, G) provide strong support for our hypothesis.

One of the most important conclusions in this study is a direct dependence of expression of the subset of complex I transcripts on Cerox1 function, therefore a very careful analysis of this observation is needed before moving to further characterization (involvement of microRNAs) and conclusions. As the phenotypic effects are very weak, it is important for the reader be convinced that there is indeed significant evidence substantiating authors' claim.

Whilst we agree with the reviewer that changes in abundance of complex I subunit mRNA transcripts are moderate, the effects on protein level (Figure 2D), cellular respiration (Figure 3A,B,C,D) and its cellular protective effect (Figure 4A, B, C) are neither weak nor insignificant. For example, *Cerox1* overexpression results in a significant 85% increase in basal respiration, and a 107% increase in ATP-linked respiration. The revised manuscript now includes a new modality – metabolomics – with which we investigated metabolite changes in *Cerox1* overexpression cells. Ten metabolites were shown to have significant effects after multiple testing (Figure 2E).

Below are specific comments with suggestions for additional experiments:Firstly, the authors claim that they provide evidence from reciprocal experiments (downregulation and overexpression of Cerox1) that Cerox1 modulates expression of a subsets of complex I subunits, however, evidence coming from downregulation experiments is weak:– Only one shRNA, moderately downregulating Cerox1 expression (by 50%), is used. It is standard practice to use at least two shRNA to convincingly show that the effect is genuine and to avoid off-target problem.

Thank you for this suggestion. From an additional six shRNAs that we screened (see Essential Revisions above), an shRNA was identified that also knocks down *Cerox1* and yields – albeit to a lesser extent – changes that are largely concordant with those observed with the most efficacious shRNA (Figure 2—figure supplement 1A, D).

– The authors observe a very mild effect (in any, as the p values are not shown, Figure 2—figure supplement 1E) of shRNA knockdown on transcripts of complex I subunits, but the effect on oxygen consumption is very pronounced. It is a quite puzzling observation. Normally, it is difficult to observe any effect on OXPHOS function, unless the mitochondrial complexes (transcripts and steady-state levels of the proteins) are substantially affected. What is even more puzzling is the fact that Seahorse analysis are done 12h after shRNA transfection.

This misunderstanding stems from an error in our previous manuscript for which we apologise. As we now clarify in the revised Materials and methods, cells were seeded before being transfected 24hrs later. Seahorse analysis was then performed 24-36hrs post-transfection.

The steady state-levels of the mitochondrial complexes are high and their half-lives long. To my knowledge, there are no studies showing a dramatic effect on oxygen consumption after such a short time that would depend on the modulation of the levels of OXPHOS transcripts. In fact, it is a routine procedure in case of transfection with RNAi that would affect mitochondrial gene expression, to measure the levels and function of the OXPHOS complexes at least 3 or 6 days after RNAi treatment, when the effect starts being visible (for example Antonicka et al. EMBO Reports, 2017 doi: 10.15252/embr.201643391 or Richter et al. EMBO, DOI 10.1038/emboj.2010.14)

We agree that the steady state levels of the mitochondrial complexes are known to be high and their half-lives long (Figure 2—figure supplement 1G, H). While some experimental systems utilise repeat transfections of siRNAs at timepoint 0 and day 3 (as in the Antonicka et al. paper referenced by the reviewer), other groups have also observed changes in oxygen consumption 24hrs post-transfection assayed using a Seahorse Bioanalyzer. For example, Drp1 was silenced in mouse neuroblastoma cells and assayed 24hrs post-transfection (Manczak et al., 2019; https://doi.org/10.1093/hmg/ddy335). These researchers also observed moderate changes in mitochondrial subunit fold changes (Manczak et al., 2019 – table 1), and observed a significant decrease in maximum uncoupled respiration at 24hrs. In addition, because the expression level of *Cerox1* co-ordinately and post-transcriptionally regulates at least 12 nuclear encoded mitochondrial complex I subunit transcripts, the large effect on oxygen consumption that we observe 24-36hrs post-transfection is possibly due to the change in abundance of multiple subunits contributing to the overall phenotype.

– Since there is such a dramatic effect on O2 consumption and almost no effect on transcripts level, to show a direct correlation and exclude a possibility of indirect effect, it is important to measure (for example by WB or quantitative mass spectrometry) the steady state levels of the complex I subunits after shRNA treatment.

While we agree that the change in transcript abundance is not dramatic, we note that these transcripts are highly expressed and very stable (Figure 2—figure supplement 1H) and therefore that a small-to-moderate fold change in transcript abundance equates to a persistent, large change in the number of transcripts per cell available for translation. Whilst an experiment utilising quantitative mass spectrometry is of substantial interest to us, unfortunately it was not possible to deliver quantitative proteomics data in the limited, 2 month, period available to us for resubmission. In particular, our prior attempts to culture cell lines of interest in SILAC medium for more than one passage resulted in cell death, which falls well short of the minimum recommendation of 5 doublings in SILAC medium to ensure >97% label incorporation (Deng, et al., 2019;

https://doi.org/10.1002/cpps.74).

We addressed the effect on complex I protein levels first from the overexpression condition (Figure 2D) mirroring the significant changes in transcript expression observed for this condition from the microarray experiment. Then, with *Cerox1* knock-down (sh92) samples and after sample normalisation, we observed a significant decrease in amounts of NDUFS1 and NDUFS3 protein (see Author response image 5) as expected from the quantitative PCR gene expression data (Figure 2—figure supplement 1E). Biochemical readouts of specific enzyme activity (Figures 3A, C) and cellular respiration (Figures 3B, D) demonstrated significant and substantial changes in cellular metabolism consistent with these protein level changes.

**Author response image 5. respfig5:** 

Apart from the knockdown experiments, further validations are also needed to show a direct correlation between effect of Cerox1 overexpression on complex I transcripts and increased OXPHOS performance:

Figures 2B, 2C, 3A, 3B, 3C, 3D show direct correlations between wildtype *Cerox1* levels and a) complex I transcripts and b) OXPHOS activity. In addition, data in Figure 5A indicates that *Cerox1* mediates its effect via a miRNA, because *Cerox1* does not affect gene expression of complex I transcripts in Dicer negative embryonic stem cells. Furthermore, Figure 5C, 5D, 6E, 6F show that the targeted ablation of the miR-488-3p binding site (MRE) in *Cerox1* abolishes its wildtype effect on the expression of 12 complex I transcripts, and on mitochondrial complex enzyme activities. Finally, overexpression of miR-488-3p was demonstrated to significantly decrease expression of 11/12 *Cerox1* sensitive complex I subunit transcripts (Figure 6A), and *Cerox1* and 11/12 complex I subunit transcripts were demonstrated to interact directly with miR-488-3p (Figure 6D, 6G). Taken together, these findings strongly support a direct role of *Cerox1* and miR-488-3p in the post-transcriptional regulation of a subset of OXPHOS transcripts.

– The qPCR validation of the expression of the OXPHOS subunits indicated by microarray shows rather weak correlation (although p values are needed to properly asses that). One could also argue that in the same time when complex I subunits are upregulated, components of complex IV are significantly downregulated (Figure 2—figure supplement 2D). However, it is activity of complex IV that is most significantly upregulated (Figure 3A).

We would like to thank the reviewer for drawing our attention to this oversight. The level of significance has now been added to Figure 2—figure supplement 1D and this indicates that 5 of the 7 complex I subunits, and 1 assembly factor are significantly upregulated. One complex III transcript is significantly downregulated (*Uqcr10*) and of the five complex IV subunits assayed by qPCR, two (*Cox6a1* and *Cox7b2*) are significantly upregulated, whilst *Cox7a2* is significantly downregulated.

– Importantly the western blot results presented in Figure 2D do not show convincingly upregulation of the investigated proteins and the quantifications presented are not reflected in shown WB blots. The quality of the part of the blot showing overexpression is poor (uneven loading -tubulin signal and diffused bands that are difficult to quantify). It would recommended to present additional blots with consistent results. Also, using other antibodies against components of complex I would be helpful (probably in Hek cell model as not many are validated in murine cells).

These experiments have been replicated and improved images and data added to Figure 2D (see Essential Revisions above). We also attempted to use two antibodies against NDUFS6 (ab230481 and PA5-19238), because the *Ndufs6* transcript showed significant gene expression changes in both overexpression and knockdown conditions. Unfortunately application of these particular antibodies has not been successful. Furthermore, by measuring gene expression changes of complex I core subunits in HEK293T cells overexpressing human *CEROX1* we obtained no evidence that the increase in complex I activity observed in these cells is associated with changes in human *Ndufs1* and *Ndufs3* transcript levels (see below).

**Author response image 6. respfig6:** 

Overexpression of *CEROX1* in HEK293T cells results in no significant changes in the expression of the core mitochondrial complex I subunits. Grey = control, Blue = overexpression. Error bars s.e.m, n = 3.

– One of the surprising results that has not been further investigated is a remarkable effect of Cerox1 overexpression on cell growth. Why would a slight upregulation of OXPHOS components lead to dramatic growth retardation? There are many studies showing that boosting OXPHOS is beneficial for the cells. Considering that microarray results shown upregulation of more than 200 transcript, the effect on cell growth can be caused by another pathway, which in the same time can indirectly affect mitochondrial performance, hence upregulation of the OXPHOS function (not necessarily directly dependent on OXPHOS transcript elevation).

We agree that the observed change in cell growth was unexpected. Nevertheless, we decided not to investigate this particular observation at greater depth owing to the large number of factors that control the timing of the cell cycle, including many that will be downstream, and not direct targets, of *Cerox1*. Our evidence from the direct binding of *Cerox1* (and complex I subunit transcripts) to miR-488, and the abolition of the effects of this binding following targeted mutation of the miR-488 binding site, indicates that the complex I subunit transcript levels measured by microarrays reflect direct effects of binding.